



# GHRSAT: the first global hourly dataset of all-sky remotely sensed estimates of surface air temperature

Zhenwei Zhang[1,2,3], Ke Wang[1], Chen Liang[4], Zihan You[1], Zishang Yi[1]

[1] School of Remote Sensing and Geomatics Engineering, Nanjing University of Information Science and Technology,
Nanjing, 210044, China
[2] Technology Innovation Centre for Integration Applications in Remote Sensing and Navigation, Ministry of Natural
Resources, Nanjing, 210044, China
[3] Jiangsu Province Engineering Research Centre of Collaborative Navigation / Positioning and Smart Application, Nanjing,
210044, China
[4] Beijing Key Laboratory of Urban Spatial Information Engineering, Beijing Institute of Surveying and Mapping, Beijing,
100038, China

*Correspondence to*: Zhenwei Zhang (zhangw@nuist.edu.cn)

**Abstract.** Spatially continuous surface air temperature (SAT) is critically important for a wide range of fields such as eco-environmental assessments and hydrology. Remotely sensed estimation models based on satellite-derived thermal infrared data provides a structurally different approach for reconstructing SAT compared to spatial interpolation of ground observations of SAT and numerical modelling, which are mainly limited by the coverage of stations and coarse spatial resolutions, respectively. However, the data products of remotely sensed estimates of SAT developed in previous studies are only available at daily or monthly resolutions, and are primarily restricted for local regions. In this study, we generated the first hourly dataset (GHRSAT) of all-sky remotely sensed SAT estimates for the global land areas except Antarctica between 2011 and 2023. The hourly estimates in GHRSAT were reconstructed from land surface temperature using the hybrid estimation models that integrate random forest (RK) models and kriging techniques. The hybrid models were developed for different regions on a monthly basis. We adopted ordinary kriging (OK) and fixed rank kriging (FRK) in the modelling of the site residuals from the RF models for regions with low-density and high-density stations, respectively. Our results show that the hybrid models for generating GHRSAT have the predictive performance between 1.48 °C to 2.28 °C in overall cross-validation RMSE. The mean RMSE for estimating hourly SAT can be significantly reduced by 0.18–0.41 °C by the hybrid models compared to the RF models. We analyzed the variability in the predictive errors of estimating hourly SAT across regions, months and sites. The variability is apparently decreased when using the hybrid models. We found that the RF models are less sensitive to the parameter tuning of the RF models, which greatly impacts the hybrid models. Improving the RF models by parameter tuning can drastically improve the hybrid models based on the RF models. Additionally, we found the performance difference between OK and FRK in developing the hybrid models for regions with large amounts of stations is slight with the mean RMSE of 0.05 °C. In summary, the scheme of the hybrid models can result in satisfactorily higher performance for estimating SAT, and has the general practicability of applying to regions at various scales. The GHRSAT dataset is publicly available at http://doi.org/10.11888/RemoteSen.tpdc.301540 (Zhang, 2024).



## 1 Introduction

Surface air temperature (SAT) at the height of about 1.5 meters above the earth's surface is an essential variable of surface meteorological observation. Reconstructing spatially continuous SAT is of great significance for a variety of fields such as assessments of epidemiological (Kloog et al., 2015; Schuster et al., 2014) and eco-environmental issues (Pichierri et al., 2012; Venter et al., 2020). Spatially explicit SAT can be reconstructed by interpolating the observations of SAT from ground stations, which is only suitable for the areas with high coverage of stations (Benali et al., 2012; Vancutsem et al., 2010). The

interpolating of in situ SAT observations has been performed only considering the spatial auto-correlation between SAT at different locations (Rohde and Hausfather, 2020), or by developing regression frameworks that incorporate the auxiliary variables for locations and topography (He et al., 2022; Qin et al., 2022b). In addition, numerical models of atmospheric dynamic processes combined with data assimilation systems have been applied to generate the reanalysis datasets that contains plentiful spatial variables for atmospheric states and surface properties, such as MERRA-2 (Gelaro et al., 2017),

ERA5 (Hersbach et al., 2020) and GLDAS (Rodell et al., 2004). However, the simulated variables of SAT in reanalysis datasets are subject to large uncertainties and available at very coarse spatial resolutions, although the temporal frequency of the variables is high.

Developing the estimation models based on remotely sensed thermal-infrared (TIR) observations provides a structurally different approach for reconstructing spatially continuous SAT. Land surface temperature (LST) retrieved from the TIR

observations of MODIS onboard the Terra and Aqua satellites has been substantially applied in developing the models for estimating daily (Huerta et al., 2023; Nikolaou et al., 2023; Yoo et al., 2018) or monthly SAT (Gao et al., 2021; Qin et al., 2023b; Yao et al., 2020, 2023). Terra and Aqua have been continuously operated in orbits for more than two decades and provide long-term global LST data products for various application areas. The LST retrievals derived from other polar satellites such as EUMETSAT's Metop and NOAA's JPSS have also been attempted in estimating daily SAT (Zhang et al.,

2024; Zhang and Du, 2022b). TIR sensors mounted on geostationary satellites are capable of scanning the earth surface at very high frequencies. It has become an increasingly important research direction in recent years to estimate SAT at high temporal resolutions based on the LST retrievals from geostationary satellites, such as the MSG satellites operated by EUMETSAT (Lazzarini et al., 2014; Meyer et al., 2019; Zhou et al., 2020b), NOAA's GOES-R satellites (Hrisko et al., 2020; Zhang and Du, 2022a), and China's FY-4 (Liu et al., 2023, 2024; Zhang et al., 2023). The models developed in previous

studies for estimating SAT primarily build on the statistical connection between truly observed SAT and the influencing covariates including LST and other auxiliary environmental variables. Various types of statistical methods such as multiple regression (Benali et al., 2012; Kloog et al., 2014; Rosenfeld et al., 2017), spatial regression (Kilibarda et al., 2014; Li et al., 2018; Nikoloudakis et al., 2020; Zhang et al., 2022c) and machine learning algorithms (Cho et al., 2020; dos Santos, 2020; Shen et al., 2020; Venter et al., 2020; Zhang et al., 2016) have been adopted in developing the estimation models for SAT.

The estimation models based on remotely sensed LST can reconstruct spatially continuous SAT with fine-scale structures. Furthermore, the estimates in the reconstructed SAT are directly constrained by LST retrievals, and indirectly constrained by



the TIR radiation observations from spaceborne satellites. However, there are massive retrieval gaps in TIR LST data because of the contamination of clouds on TIR observations. The previous studies for estimating SAT based on TIR LST are primarily limited to clear-sky areas. Although the study by Zhang and Du (2022b) has attempted to develop the merging

framework based on the LST retrievals from multiple polar satellites to improve the spatial coverage of daily SAT estimates, it remains a challenge to reconstructing all-sky SAT from LST retrievals. Several attempts have been made to estimate all-sky SAT by firstly reconstructing spatially complete LST using spatio-temporal interpolation methods (Chen et al., 2021; Gutiérrez-Avila et al., 2021; Yao et al., 2023; Zheng et al., 2022), or by integrating LST with reanalysis data into the estimation models for SAT (Fang et al., 2022; Qin et al., 2022a; Zhang et al., 2021). Reconstruction of LST is also important

research field of quantitative TIR remote sensing (Li et al., 2023; Wu et al., 2021)for tackling the issues of retrieval gaps, spatio-temporal resolutions and inconsistency (Duan et al., 2017; Martins et al., 2019; Shwetha and Kumar, 2016; Zhao et al., 2019), which has great implications for the remotely sensed estimation of SAT. There are the studies performed to reconstruct all-weather LST data products based on the TIR LST retrievals from polar satellites (Shiff et al., 2021; Zhang et al., 2022b) or geostationary satellites (Jia et al., 2021, 2023; Martins et al., 2019). Few studies have attempted to develop the

models based on reconstructed LST for estimating all-sky SAT (Wang et al., 2022; Zhang et al., 2022c).

**Table 1. Summary of the publicly available products of all-sky remotely sensed SAT estimates.**

| Study | Spatial Coverage | Temporal Coverage | Spatial Resolution | Temporal Resolution | Journal Abbr. |
|---|---|---|---|---|---|
| Wang et al. (2024) | China | 2003–2022 | 1 km | Daily | Sci. Data |
| Yao et al. (2023) | Global | 2001–2020 | 1 km | Monthly | Remote Sens. Environ. |
| Qin et al. (2023a) | Tibetan | 2002–2020 | 1 km | Monthly | Earth Syst. Sci. Data |
| Nielsen et al. (2023) | Antarctic | 2003–2021 | 1 km | Daily | Sci. Data |
| Zhang et al. (2022b) | Global | 2003–2020 | 1 km | Daily | Earth Syst. Sci. Data |
| Fang et al. (2022) | China | 1979–2020 | ~10 km | Daily | Earth Syst. Sci. Data |
| Zhang et al. (2021) | Tibetan | 1980–2014 | 1 km | Daily | Int. J. Appl. Earth Obs. Geoinf. |
| Chen et al. (2021) | China | 2003–2019 | 1 km | Daily | Earth Syst. Sci. Data |
| Hooker et al. (2018) | Global | 2002–2016 | ~5 km | Monthly | Sci. Data |

Compared to the large number of studies aimed at developing various models for estimating SAT at different scales, there are very limited studies performed to generate data products of remotely sensed SAT estimates based on TIR LST data. The

studies for developing publicly available products of all-sky SAT estimates are summarized in Table 1, and there are no





studies for developing the estimated SAT products at high temporal resolutions. The daily products of remotely sensed SAT developed by Zhang et al. (2022a) and Wang et al. (2024) were generated from a reconstructed daily LST product (Zhang et al., 2022b), while Chen et al. (2021) and Yao et al. (2023) integrated seamless reconstruction of MODIS LST data with statistical estimation models in the developing of the products of daily SAT in China and monthly SAT in global land areas,

respectively. The monthly LST datasets composited from daily MODIS LST retrievals have nearly complete spatial coverage, and have also been utilized to develop seamless products of daily SAT (Huerta et al., 2023) and monthly SAT (Hooker et al., 2018). However, the products of remotely sensed SAT are primarily for local-scale regions such as China and the Tibetan Plateau (Qin et al., 2023a; Zhang et al., 2021), and these products are limited to daily or monthly temporal resolutions. Furthermore, the products were generated by the estimation models only based on machine learning or spatial

regression methods. For example, random forest is utilized to developed the models for generating the products of daily SAT (Chen et al., 2021) and monthly SAT (Qin et al., 2023a), while the Cubist method was applied in developing the global monthly SAT product (Yao et al., 2023). Zhang et al. (2022a) developed the models based on spatially varying coefficient regression for generating a global daily SAT product. To improve the predictive accuracy of SAT, the modelling strategy integrating statistical learning methods and spatial models (Huerta et al., 2023; Nikoloudakis et al., 2020) can be employed

in developing the models for estimating SAT.

In this study, we developed the hybrid models for estimating hourly SAT in the global land areas, and for the first time, generated the global hourly data product (GHRSAT) of remotely sensed all-sky estimates of SAT from 2011 to 2023. The hybrid estimation models for generating GHRSAT were constructed by the two-stage modelling strategy that integrates ensemble learning with kriging techniques. The estimation models based on random forest were developed in the first stage

for estimating hourly SAT from seamless reconstructed LST. The residuals at ground stations from the models developed in the first stage were modelling by ordinary kriging for areas with limited stations and the computationally efficient method of fixed rank kriging for areas with high-density coverage of stations. We comprehensively assessed the predictive performance the hybrid models, and compared the models with the estimation models only based on random forest for estimating hourly SAT. In addition, we analyzed the impacts of learning parameters for random forest on the predictive performance of the

hybrid models, and the issues involved in the kriging modelling of large numbers of stations in the second stage of hybrid models. We expect that the remotely sensed product of SAT estimates generated by our study will provide an important data basis for eco-environmental assessments and other related fields.

## 2 Study areas and data

### 2.1 Study areas

The GHRSAT product was generated by developing the hybrid estimation models for the global land surface areas excluding the Antarctic region due to the severe scarcity of ground stations in the region. The models were developed separately for

eight task regions (Fig. 1) on the monthly basis because (i) the locally developed models have high adaptability to the areas with distinct geographical contexts; (ii) the computational burden of training the models is feasible in this work. In contrast to the study (Zhang et al., 2022b) that generated the product of global daily SAT by separately constructing the estimation

models for different landmass of continents, the task regions for developing our models were partitioned by considering the spatial unevenness of ground stations across the areas. The models trained using the samples from the stations in a region only fit the average relationships represented by the samples. If there is the distinct contrast in the density of stations across a region, the fitted relationships by the model developed for the region will be severely smoothed and biased towards the relationships represented by the samples from the locations with high-density of stations in the region. Thus, the partitioning

of the global land areas for developing the hybrid models can ensure the objective assessment of the models for the regions with low-density stations. The regions of TR-2 and TR-4 have high density of stations, primarily covering conterminous U.S. and Europe, respectively. The TR-1 and TR-6 regions located in the Arctic have very limited coverage of stations.

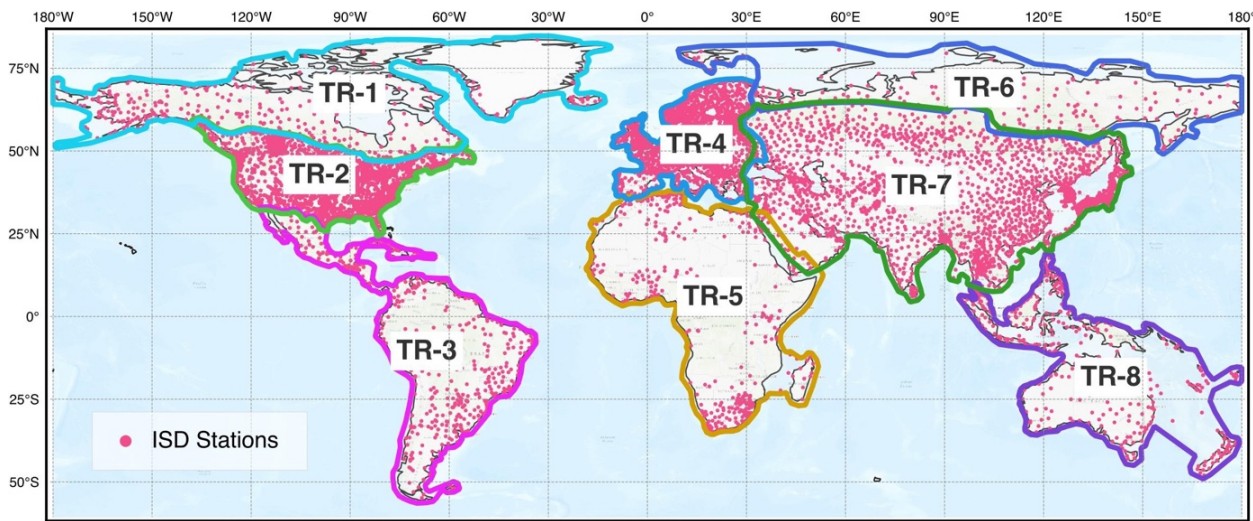

**Figure 1: Task regions (TR) defined to separately develop the estimation models for hourly SAT. The TR regions**
**are denoted by boundary lines in colours. The dot symbols represent the position of ground weather stations.**

**2.2 Ground station data**

The hourly observations of SAT at ground stations used in this study were extracted from the ISD (Integrated Surface Database) product maintained by NOAA NCEI (https://www.ncei.noaa.gov/data/global-hourly/). ISD was developed by integrating hourly and synoptic observations of surface meteorological parameters such as air temperature, pressure and

wind speed from more than 100 sources, and the records contained in ISD have been subject to strict manual and automatic quality-control measures (Smith et al., 2011). Fig. 1 shows the locations of the ground stations with at least one valid hourly record of SAT during the 2011–2023 period. The spatial density of the stations is highly imbalanced across the land areas. The areas of central North America (TR-2) and Europe (TR-4) are covered with very high-density coverage of stations, and



most of the stations in the areas have hourly continuous records of SAT (Fig. S1 and Fig. S2). In contrast, the number of the
stations with available SAT records varies greatly across the hours of the day for other task regions such as TR-7 and TR-5
(Fig. S1) because the source data of the stations integrated into ISD for these regions only contain the records of SAT at
fixed synoptic hours.

**2.3 Data for spatial covariates**

We developed the hybrid models for estimating all-sky hourly SAT based on GHA-LST, which is a spatially seamless
dataset that contains global hourly LST (http://glass.umd.edu/allsky_LST/GHA-LST/). The GHA-LST dataset was generated
by synthesizing TIR LST retrievals from geostationary satellites and polar satellites using the three-stage reconstruction
framework involving a time-evolving model, the assimilation process based on Kalman filter, and the removal of cloud
effects (Jia et al., 2023). In the development of the models for estimating SAT based on LST, various auxiliary spatial
covariates have been attempted in previous studies (Janatian et al., 2017). Although several recent studies have demonstrated
that the predictive performance of estimating SAT can be improved by incorporating the reanalysis simulated variables of
atmospheric states into the models (Shen et al., 2020; Wang et al., 2024; Zhang and Du, 2022a), it is hard to objectively
assess the models. As the simulated variables used in the models are from the reanalysis of numerical models assimilating
large numbers of observations from ground stations, it will underestimate the predictive errors of estimating SAT by the
models when trained using the samples extracted at the stations of which many have been assimilated for generating the
simulated variables. Thus, the predictive capability of estimating SAT will be confounded by the simulated variables
included in the models. In this study, to develop the models for estimating hourly SAT, we only utilized the auxiliary
variables that have been extensively validated by previous studies and for which the data are publicly available at the global
scale. Specifically, the auxiliary spatial covariates used in the hybrid models include the variables for normalized vegetation
index (NDVI), topographic elevation (ELE), longitude (LON), latitude (LAT) and hours of day (HOD) were used in the
models. The data for NDVI and ELE were extracted from MOD13A2 (https://lpdaac.usgs.gov/products/mod13a2v006/) and
GMTED2010 (https://www.usgs.gov/coastal-changes-and-impacts), respectively.

# 3 Methods

We generated the GHRSAT product by developing the estimation models for estimating all-sky hourly SAT using the hybrid
modelling strategy that integrates machine learning algorithms and kriging techniques to improve the predictive accuracy of
hourly SAT estimates. The models developed by the hybrid strategy are thereafter referred as hybrid estimation models for
brevity in this study. The models for estimating SAT developed in previous studies are primarily based on machine learning
algorithms (Chen et al., 2021; Meyer et al., 2019; Yoo et al., 2018) or spatial regression methods (Lu et al., 2018; Zhang et
al., 2022c; Zhang and Du, 2019). Our hybrid models were constructed in two stages. In the first stage, the estimation models
based on random forest were developed to represent the connection between SAT and spatial covariates using the samples





extracted at ground stations. The residuals at ground stations computed from the first-stage models were modelled in the second stage using kriging methods to represent the spatial structures in the residuals. Both the models developed in the first stage and the hybrid models were independently cross-validated. The overall framework for developing the hybrid models is schematically shown in Fig. 2, and involves two major processes: (i) processing all input data to obtain the samples extracted at ground stations, and the uniformly gridded spatial covariates with the 5-km resolution for each hour point in the 2011–

2023 period; (ii) developing the hybrid models for each task region to generate the hourly estimates of SAT in the region.

**Figure 2: Overall schematic diagram for the modelling of hourly SAT using the hybrid estimation models.**



## 3.1 Data processing

We processed the data for spatial covariates for each hour in the period of 2011–2023 into the regular grid with a resolution of 0.05° in which the hourly LST from GHR-LST is organized. The hourly estimates of SAT in the GHRSAT product were also reconstructed using the grid. The data fields for NDVI in the data files of MOD13A2 and the elevation variable (ELE) from GMTED2010 were reprojected and resampled into the regular grid. The gridded layers of LON and LAT were directly generated from the central geographic coordinates of the grid boxes in the grid, and the grid layers of HOD contain constant values for all grid boxes representing each hour of the day. We first extracted the records of air temperature measured at the stations in the task regions (Fig. 1) for the 2011–2023 period from the ISD dataset. The extracted records were further processed by two steps: (i) removing the records with missing observations or quality-control issues; (ii) filtering the records observed within a time window of 15 minutes centred at each hour, and the records within the window were aggregated to compute the average SAT for each hour and each station. The processed records of SAT at stations were spatio-temporally matched with the stacks of gridded covariates for each hour to obtain the samples used for training the models for estimating hourly SAT. We obtained about 0.9 billion matched samples at ground stations for all 8 task regions in the 2011–2023 period, and the average number of samples for each calendar month is about 5.9 million.

## 3.2 Hybrid estimation models

The estimated hourly SAT from hybrid models for the grid cell $s_i$ is modelled as $SAT(s_i) = f(X(s_i)) + w(s_i)$, where $f(\cdot)$ is a statistical model that estimates hourly SAT using the spatial covariates $X$ at the cell $s_i$; $w(\cdot)$ is a random field with spatially auto-correlated structures for modelling the residuals in the statistical model. The hybrid estimation models were constructed successively in two stages (Fig. 2). We adopted random forest (RF) in the first stage for the modelling of $f(\cdot)$ using spatial covariates to estimate hourly SAT, which is formulized as follows:

$$SAT_0 = f_{RF}(x_0, x_{LST}, x_{NDVI}, x_{ELE}, x_{LON}, x_{LAT}, x_{HOD}; \boldsymbol{\beta}) \qquad (1)$$

The $\boldsymbol{\beta}$ in Eq. (1) denotes the parameters to be tuned for random forest. Random forest is a highly efficient ensemble learning algorithm with the ability to model complex non-linear relationships, and has been widely used in the remotely sensed SAT estimation (Chen et al., 2021; Venter et al., 2020; Yoo et al., 2018) and other fields of remote sensing (Belgiu and Drăguţ, 2016; Wei et al., 2019, 2021; Zhao et al., 2019). The algorithm is less prone to model overfitting with the advantage of insensitivity to the tuning of parameters (Meyer et al., 2018). Specifically, the models for $f_{RF}$ were developed for each task region on the monthly basis, and are referred as RF in this text. The RF models constructed in the first stage of the hybrid models for generating the GHRSAT product were trained using the parameters optimally determined from the tuning grid of the four core parameters (Table S1).

The residuals at ground stations were computed as the difference between the observed SAT and the predicted $SAT_0$ by the first-stage RF models. The residuals for each hour were modelled by kriging techniques to generate the kriged residual $w(s_i)$





in each grid cell $s_i$ of the regular grid with a resolution of 0.05°. Two types of kriging techniques including ordinary kriging

(OK) and fixed rank kriging (FRK) were utilized in the second stage for different task regions. The hybrid models developed in this study specifically refer to the integration of the modelling performed in two stages, and are referred as RF-KR. The two types of hybrid models including RF-OK and RF-FRK only differ in the techniques used for the second stage of the hybrid models. Conventional kriging techniques are computationally intensive due to the solving of the inversion of matrix, and especially, it is impractical to apply ordinary kriging in the second stage of the hybrid models for each hour between

2011 and 2023 for the task regions with large numbers of ground stations. Thus, the RF-OK hybrid models were developed for the regions including TR-1, TR-3, TR-5, TR-6 and TR-8, while the RF-FRK hybrid models utilizing computationally efficient FRK were developed for the regions with large numbers of stations, including TR-2, TR-4 and TR-7.

Ordinary kriging is the best linear unbiased interpolation method that constructs the gridded distribution for a spatial field from point samples of the field at different stations. The kriged residual at each grid cell is computed as the linear weighted

average of the residuals at the stations: $w(s_i) = \boldsymbol{\lambda}^T \boldsymbol{\varepsilon}$. The $\boldsymbol{\lambda} = [\lambda_1, \lambda_2, \cdots, \lambda_N]^T$ represents the weights chosen for each station to minimize the error variance by solution of the set of equations:

$$\sum_{i=1}^N \lambda_i \gamma(s_i - s_j) + \psi(s_0) = \gamma(s_j - s_0), j = 1,2, \cdots, N; \sum_{i=1}^N \lambda_i = 1 \tag{2}$$

Here $\boldsymbol{\varepsilon} = [\varepsilon_1, \varepsilon_2, \cdots, \varepsilon_N]^T$ is the vector of the residuals at the stations; $\gamma(s_i - s_j)$ denotes the semivariance between stations $s_i$ and $s_j$; $\gamma(s_j - s_0)$ represents the semivariance between a station $s_j$ and the centre of the grid cell $s_0$ in which a kriged

value will be computed. There are various types of variogram models for characterizing the semivariance. The spherical model was used in the OK equations, and is expressed in Eq. (4), where the $c_0$ is the nugget term; the $h$ is the spatial lag distance between two spatial points, and $c$ denotes the spatially correlated variance. The parameters of the spherical model were fitted from the residuals $\boldsymbol{\varepsilon}$.

$$\gamma_{Sph}(h) = c_0 + c \left\{ {3h}/{2r} - {h^3}/{2r^3} \right\} \tag{3}$$

In the RF-FRK hybrid models for regions with large numbers of stations, the highly efficient kriging method FRK was used for the kriging modelling in the second stage. The residuals at stations are regarded as a sampling from a spatial random field $w(s)$ with dependent structures. The random field is represented as the weighted combination of a fixed number of spatial basis functions: $w(s) = \sum_{l=1}^r \phi_l(s)\eta_l + \epsilon(s)$, where $\phi_l(s)$ is one spatial basis function with a specific parametric form; the weight for the function is denoted by a random effect variable $\eta_l$; $\epsilon(s)$ is a white noise term. The representation of the field

is expressed in the following matrix form:

$$w(s) = \boldsymbol{\eta}^t \boldsymbol{\phi}(s) + \epsilon(s) \tag{4}$$

$$\boldsymbol{\eta} = (\eta_1, \dots, \eta_r)^T \tag{5}$$

$$\boldsymbol{\phi}(s) = \left( \phi_1(s), \dots, \phi_r(s) \right)^T \tag{6}$$





When the r-dimensional random vector $\boldsymbol{\eta}$ is a normal distribution with zero means and the covariance matrix $\boldsymbol{K}_{r \times r}$, the
spatial covariance function for the field $w(s)$ is represented as $C_\lambda(s) = \boldsymbol{\phi}^t(s)\boldsymbol{K}\boldsymbol{\phi}(s) + \tau^2\boldsymbol{I}$. For the $N$ samples at stations,
the values of the basis functions at the stations are formed as the matrix $\boldsymbol{S}_{N \times r}$, and the covariance function for the residuals at
the $N$ stations can be formed by the covariance matrix $\boldsymbol{\Sigma}_\lambda = \boldsymbol{SKS}^t + \tau^2\boldsymbol{I}$. The matrix inversion in the FRK kriging method is
performed for $\boldsymbol{\Sigma}_\lambda$, which involves the computation of inverting the r-dimensional matrix $\boldsymbol{K}_{r \times r}$. In the application of the FRK
method, the number of spatial basis functions $r$ is very small, and the number of stations $N \gg r$. Thus, the computation of
FRK is very efficient for kriging large numbers of stations. In addition to the efficiency of FRK, the method is capable of
modelling spatially non-stationarity in the residuals at stations by combining the spatial basis function chosen at different
scales (Zammit-Mangion et al., 2018). More detailed theoretical and practical descriptions of FRK can be referred to Cressie
et al. (2008) and Kang et al. (2011).

### 3.3 Model training and validation

The hybrid models for generating GHRSAT were developed on the monthly basis for each task region, and thus there are
156 different RF models to be trained in the first stage of the hybrid models for each task region across the months between
2011 and 2023. The RF models developed in the first stage were tuned for the core parameters (Table S1), which were
determined from an exploratory cross-validation of the models. As such, the RF model or the RF-KR model for each task
region and each month was cross-validated separately using a total of 72 different sets of parameters. The validation of a
255    hybrid model was performed in the following steps that include: (i) the samples used for model training were randomly
partitioned into 10 folds with equal sizes; (ii) the RF model in the first stage was trained using the training set that includes 9
folds of samples, and the remaining one fold of samples was used as the testing set; (iii) residuals for the training samples
were computed from the trained RF model; (iv) the residuals at stations for each hour were modelled using kriging methods
in the second stage; (v) the predicted SAT by the RF model in the first stage for the testing samples was computed, and the
260    predicted SAT by the hybrid model for the testing samples was computed by adding the predicted SAT in the first stage to
the kriged residuals at the site locations of the testing samples in the second stage; (vi) the previous steps were repeated until
each fold of samples had been used as the testing set.

The predictive performance of the hybrid models or the first-stage RF models is assessed based on the difference between
the predicted SAT and the truly observed SAT for the testing samples. Two statistical metrics including root mean squared-
265    error (RMSE) and mean absolute error (MAE) were used for measuring the performance of the models for estimating hourly
SAT. The RMSE and MAE are computed by Eq. (7) and Eq. (8), respectively.

$$RMSE = \sqrt{1/N \sum_{i=1}^N (y_i^{pred} - y_i^{obs})^2} \qquad (7)$$

$$MAE = \frac{1}{N}\sum_{i=1}^N |y_i^{pred} - y_i^{obs}| \qquad (8)$$





Here $y_i^{pred}$ and $y_i^{obs}$ are the predicted SAT and observed SAT for one testing sample, respectively; the $N$ is the number of testing samples used in the computation of RMSE or MAE.

Two types of hybrid models including RF-OK and RF-FRK were developed for different task regions. The RF-FRK models were designed for the regions of TR-2, TR-4 and TR-7 due to the computation burden of ordinary kriging for large numbers of stations. Although the feasibility of implementing the hybrid modelling strategy for estimating SAT can be ensured by applying computing efficient kriging methods such as FRK in the second stage, the impacts of different kriging techniques on the predictive performance of the hybrid models should be assessed. We analyzed the impacts of ordinary kriging and fixed rank kriging on the performance of the hybrid models for estimating hourly SAT by comparing the cross-validated results of the RF-OK models with the RF-FRK models across the months in 2020 for TR-2, TR-4 and TR-7, and these models were all based on the same RF models trained for each month and each region.

## 4 Results and discussion

### 4.1 Overall performance of estimation models

Global all-sky hourly estimates of SAT in GHRSAT were generated by merging the SAT estimates predicted by the hybrid models for all task regions. Fig. 3 shows the distribution of the SAT estimates and the distribution of kriged residuals in the second stage of the hybrid models for the hour 0 of 1 January 2016. The kriged residuals indicate that the RF models have apparent spatially structured prediction errors across the land areas. The RF models generally overestimate hourly SAT in the coastal areas of northern Asia, southeast Australia and northwest China, and the areas such as northern Asia and eastern Europe are prone to be underestimated by the RF models. The hybrid models can reduce the spatially structured errors in the RF models by further modelling the residuals at stations from the RF models using kriging techniques.

The hybrid models (RF-KR) for generating the GHRSAT product were developed for each task region across the months in the 2011–2023 period. As such, there are 156 different hybrid models developed and cross-validated for each region. We computed the RMSE and MAE from the validated testing samples for each hybrid model (RF-KR) and the RF model in the first stage of RF-KR to characterize the overall predictive performance of estimating hourly SAT for each region. Fig. 4 presents the variability in the overall predictive performance of the models for different task regions. The RMSE for each cross-validated model including both RF-KR and RF developed across the 156 months for each region is shown in Fig. 5. The RF-KR hybrid models and the RF models show great variability in the predictive performance of estimating hourly SAT across different regions, and perform significantly poorer for the regions with the scarcity of ground stations such as TR-1 and TR-2 (Fig. 4). Furthermore, we can see that the hybrid models consistently achieve higher predictive performance than the RF models across the months between 2011 and 2023 for each region (Fig. 5). The averaged performance of the hybrid models for different regions ranges from 1.48 °C to 2.28 °C in RMSE, and from 1.07 °C to 1.68 °C in MAE. Compared to RF-KR, the RF models achieve lower performance with the average RMSE and MAE ranging from 1.67 °C to 2.54 °C, and

from 1.21 °C to 1.87 °C, respectively.

The RF-KR models and RF models developed for TR-1 and TR-6 are characterized by both high average predictive errors and high variability in the errors across the months. The RF-KR models for TR-1 and TR-6 across the months respectively have the RMSE between 1.46 °C and 3.12 °C with the standard deviance of 0.35 °C, and between 1.60 °C and 3.16 °C with the standard deviance of 0.42 °C. In contrast, the RF-RK models across the months for the other 6 regions have the RMSE

between 1.28 and 2.05 °C, and the standard deviance of the RMSE for the models ranges from 0.05 °C to 0.15 °C. In average, the hybrid models greatly improve the predictive performance of estimating hourly SAT for the regions by 0.18−0.41 °C in terms of RMSE relative to the RF models, and the variability in the performance of the hybrid models across the months is also reduced. The hybrid models achieve the remarkable improvement to the estimation performance by about 0.41 °C in the averaged RMSE with respect to the RF models for TR-2.

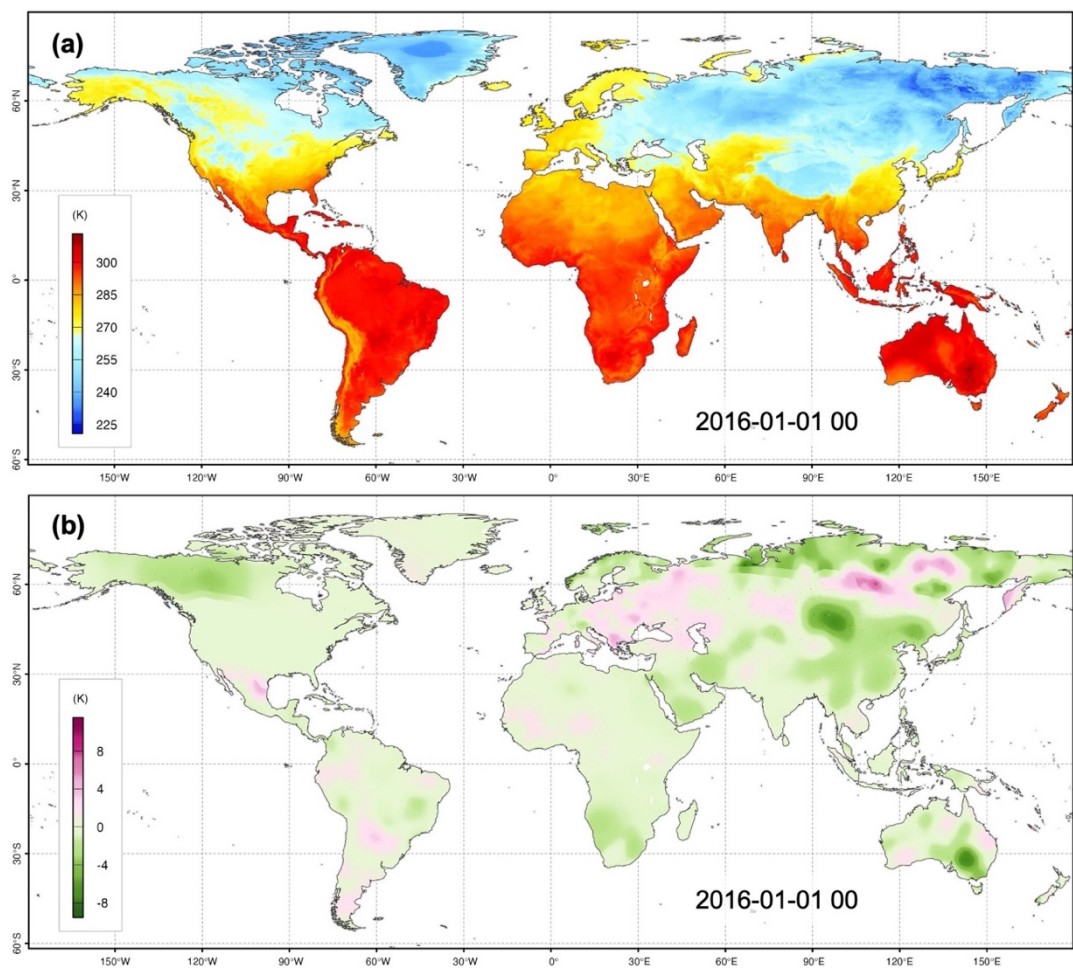

**Figure 3.** An example of the spatial distribution of hourly all-sky SAT reconstructed by hybrid models (a) and the kriged residuals generated by the residual modelling in the second stage of the hybrid models (b).


Earth System
Science
Data

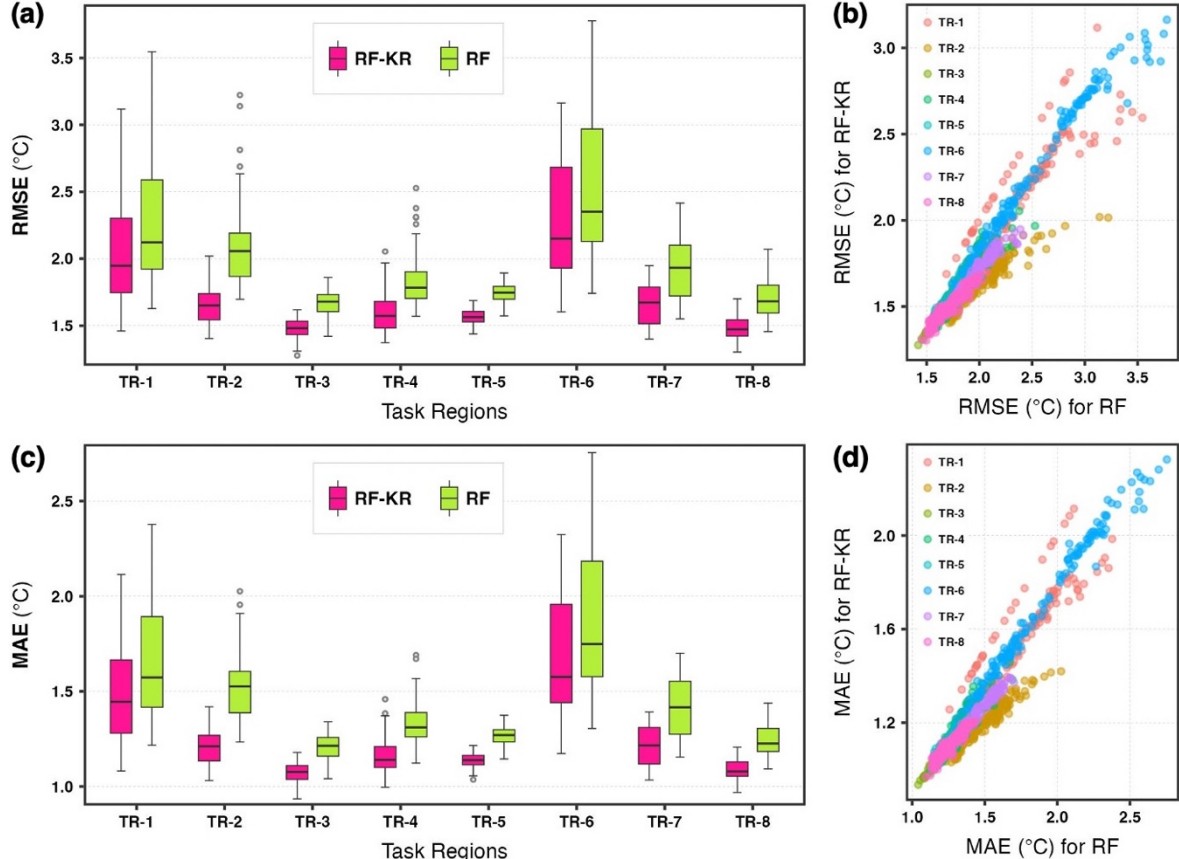

**Figure 4. Predictive performance of the hybrid estimation models developed for different task regions on the monthly basis to generate the GHRSAT product. The variability in the overall cross-validated RMSE and MAE for the hybrid models (RF-KR) in comparison to the models (RF) developed in the first stage of the hybrid models are shown in (a) and (c), respectively. The panels of (b) and (d) show the connection in the predictive performance between the RF models and the RF-KR models in terms of RMSE and MAE, respectively.**

Both the RF-KR models and the RF models developed for the same region exhibit the variability in predictive performance of estimating SAT, which is due to the impacts of the seasonality on developing the models for estimating SAT. As shown in Fig. 5, the RF-KR or RF models for the same month in different years have similar cross-validated overall RMSE. The models show greater variability in the performance with the average RMSE and MAE above 2.0 °C and 1.5 °C, respectively, for TR-1 and TR-6, which are the polar regions with very limited stations. In contrast, the models for the other six regions have the lower variability in performance across the months, and the models were trained using the samples from the stations with high coverage in these regions, especially in the regions of TR-2/4/7. The impacts of the seasonality on the estimation models for SAT have been reported in the previous studies for modelling SAT at monthly (Gao et al., 2021; Yao et al., 2020), daily (Wang et al., 2024; Zhang et al., 2022c; Zhou et al., 2020a) or hourly scales (Zhang and Du, 2022a). In general, the variability in the predictive performance of estimation models across months is characterized by high predictive errors for





winter months and lower errors in summer months. The monthly variability can be reduced to some degrees by improving
the overall performance of the estimation models, which can be achieved by incorporating more influencing spatial
covariates into the models (Wang et al., 2024; Yao et al., 2020; Zhang and Du, 2022a). In particular, Yao et al. (2022)
constructed the estimation models for daily SAT with the improvement to the models by the EPC method that specifically
corrects large prediction errors for extremely high and low ranges of SAT, resulting in high overall performance with the
variability in the performance across months greatly decreased. Although incorporating the variables for atmospheric states
from reanalysis datasets such as ERA5 into estimation models has been demonstrated to improve the overall performance
and reduce the monthly variability in the performance of estimating daily (Shen et al., 2020) or hourly SAT (Zhang and Du,
2022a), reanalysis datasets are generated by numerical models with the assimilation of large quantities of observations from
ground stations. Thus, the variables from reanalysis datasets were not included in the RF-KR models developed by this study
to guarantee the independence between the ground observations and the spatial covariates used in the models, which enables
the objective cross-validation of the models.

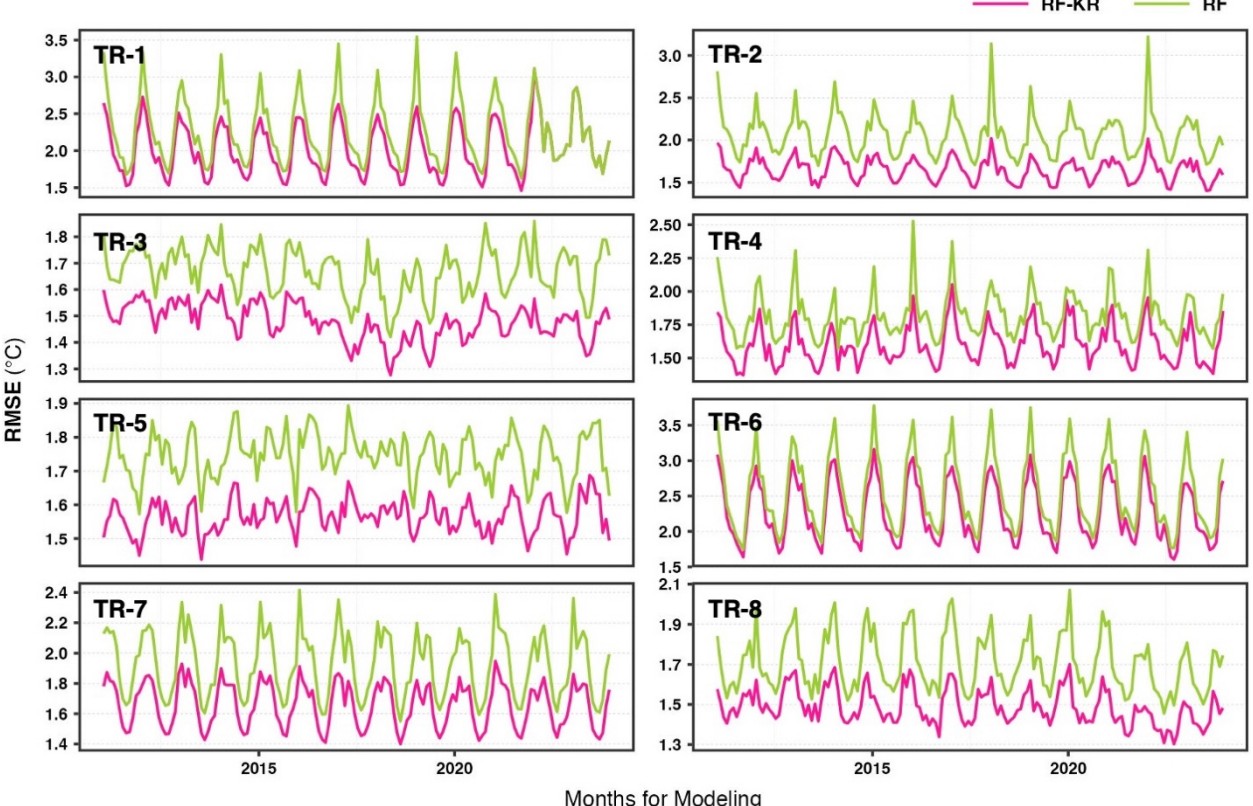

**Figure 5. Overall predictive performance (RMSE) of the hybrid estimation models (RF-KR) developed for each month between 2011 and 2023 and each region to generate the GHRSAT product. The RMSE for the RF models developed in the first stage of RF-KR is shown for the comparison with the RF-KR models.**



Therefore, the previous results justify that estimating SAT based on the hybrid models that integrate machine learning and the residual modelling by kriging methods can significantly improve the estimation accuracy of SAT. Fig. 4 and Fig. 5 show that the hybrid models achieve consistently higher predictive performance of estimating hourly SAT compared to the RF models. As shown in (b) and (d) of Fig. 4, there is the approximately linear relationship between the performance of the RF-KR models and the performance of the RF models constructed in the first stage of RF-KR, which suggests that higher overall
predictive performance of estimating SAT can be expected when improving the performance of the models constructed in the first stage of the hybrid models. However, it is unclear that the improvements to the models in the first stage impact the improvements to the overall performance of the hybrid models, which will be analyzed in the following text.

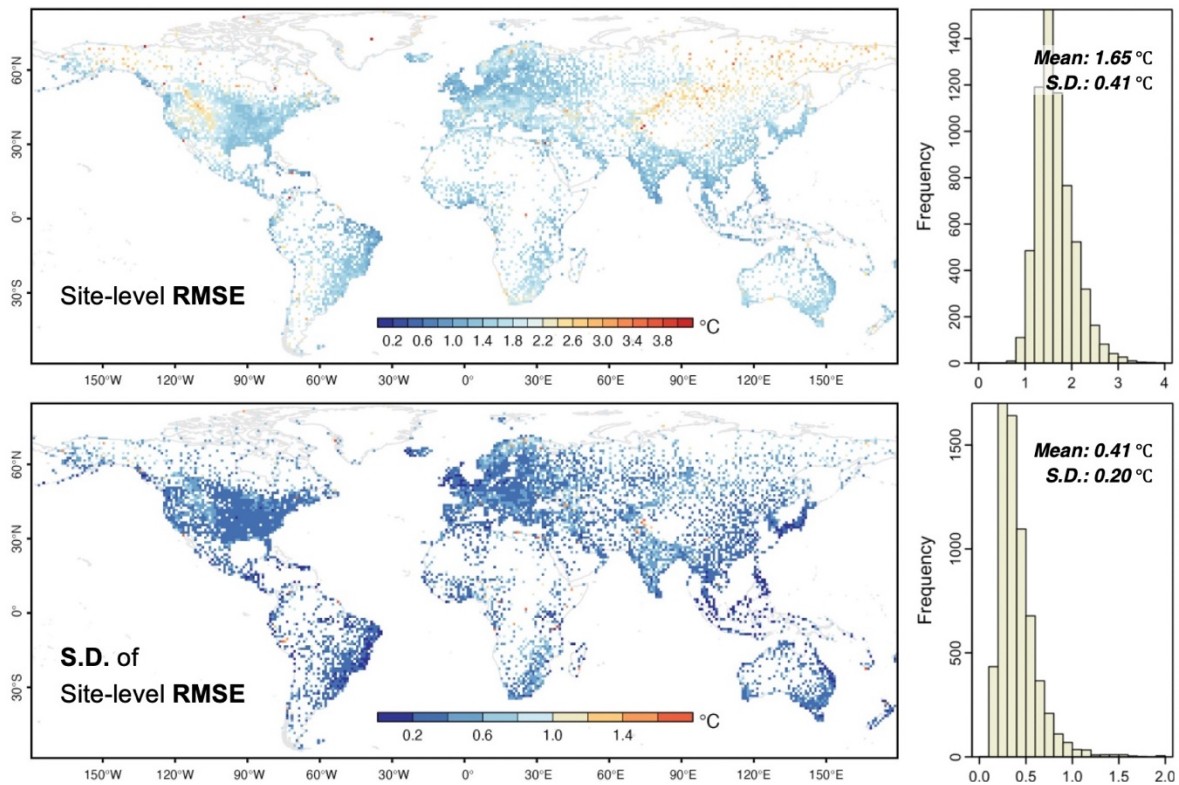

**Figure 6. Averaged site-level RMSE and its standard deviance (S.D.) for the sites within 1-degree grid boxes. The**
**site-level RMSE for each site was computed for each hybrid model developed across the months between 2011 and 2023 using cross-validated samples. The right panels show the distribution of the averaged site-level RMSE and its S.D. for all sites.**

## 4.2 Spatial analysis of model performance

We computed the site-level RMSE for a hybrid model using the cross-validated samples for each site. As such, we obtained
a sequence of site-level RMSE for one site computed from the validated samples for the hybrid models developed across the months between 2011 and 2023. The averaged site-level RMSE and its standard deviance (S.D.) were calculated from the



sequence of site-level RMSE for each site, and are shown in Fig. 6. To better visualize the prediction errors of the hybrid models for generating GHRSAT across the global land areas, the averaged site-level RMSE and its S.D. for the sites were spatially binned into 1-degree grid boxes in the figure. The predictive performance of the hybrid models across the sites in

terms of RMSE are primarily between 0.15 and 4.2 °C with the mean of 1.65 °C. The averaged site-level RMSE for most of sites is below 2.5 °C, while the sites in the mountainous regions or the regions with very limited coverage of sites have comparatively higher RMSE. The hybrid models exhibit evidently poor performance with RMSE above 2.5 °C for the sites in the high-latitude regions adjacent to the north pole (TR-1 and TR-6), the Rocky mountainous region, the Pamir Plateau, the Tibetan Plateau, and the Mongolian Plateau. These regions are subject to the scarcity of ground sites. Previous studies

have demonstrated that the estimation models for SAT generally have relatively higher prediction errors for the regions with limited coverage of sites (Chen et al., 2021; Kilibarda et al., 2014; Zhang and Du, 2022a). Furthermore, these regions are typically characterized by complex geographical environments and atmospheric dynamics, which severely affects the spatial representativeness of the samples extracted from the limited stations in these regions. It remains a challenge to develop the estimation models with reduced prediction errors for the regions with limited stations. The S.D. of the averaged site-level

RMSE across the sites as shown in Fig. 6 confirms the variability of the performance of the hybrid models for each site across the months, which is due to the impacts of seasonality on the hybrid models.

The averaged site-level RMSE for each site was also computed for the RF models developed in the first stage of the hybrid models. Fig. 7 shows the differences in the averaged site-level RMSE between the RF models and the RF-KR models across the sites. We see that nearly all sites have positive differences with the mean difference of 0.24 °C, indicating that the hybrid

models consistently improve the predictive performance of estimating hourly SAT across the sites. There is great spatial variability in the differences, and the S.D of the differences is about 0.13 °C, reflecting that spatial location is an important factor in the estimation modelling of SAT. In particular, the hybrid models greatly improve the predictive performance for the sites in the central areas of north America with the differences above 0.5 °C.

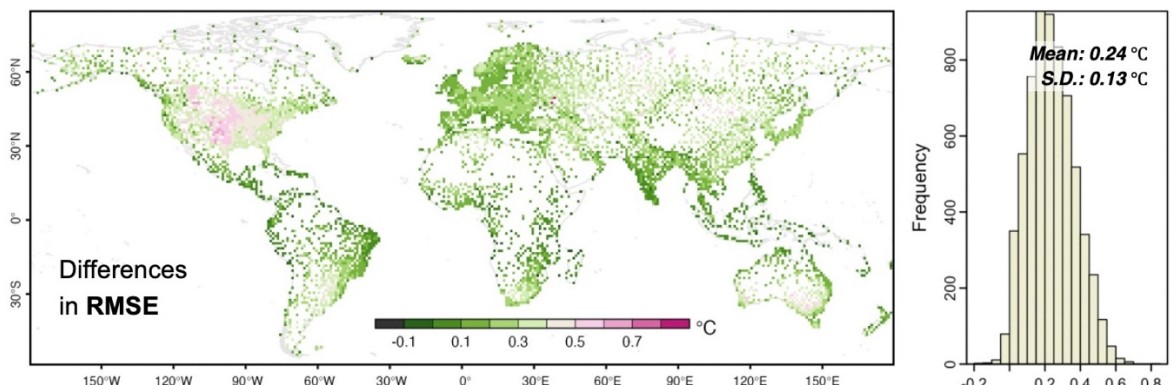

**Figure 7. Spatial distribution of the difference in the averaged site-level RMSE between the hybrid models and the RF models for the sites. The differences for the sites within 1-degree grid boxes were binned for visualization.**

We analyzed the variability in the predictive performance of the hybrid models across different elevations. Fig. 8 compares the averaged site-level RMSE for the sites against the elevations of the sites. The figure shows that the variability in both the site-level RMSE and the S.D. of the site-level RMSE is greater for the sites in low elevations than that for the sites in high elevations, which is primarily due to the inadequacy of sampling by the very limited number of sites in high elevations. The number of available sites decreases significantly with the increasing of elevations. For the sites with the elevation below 1000 m, the range of the site-level RMSE is about between 0.5 °C and 4 °C, while the range of the site-level RMSE is narrowed between about 1.2 °C and 3 °C. The mean level of the site-level RMSE for the sites at different elevations is slightly increasing with elevations, mainly because the models generally perform poorer for the high-elevation regions. In addition, the relationship between the S.D. of the site-level RMSE and elevations shows that there is higher variability in the site-level RMSE for the sites with low elevations than that with high elevations.

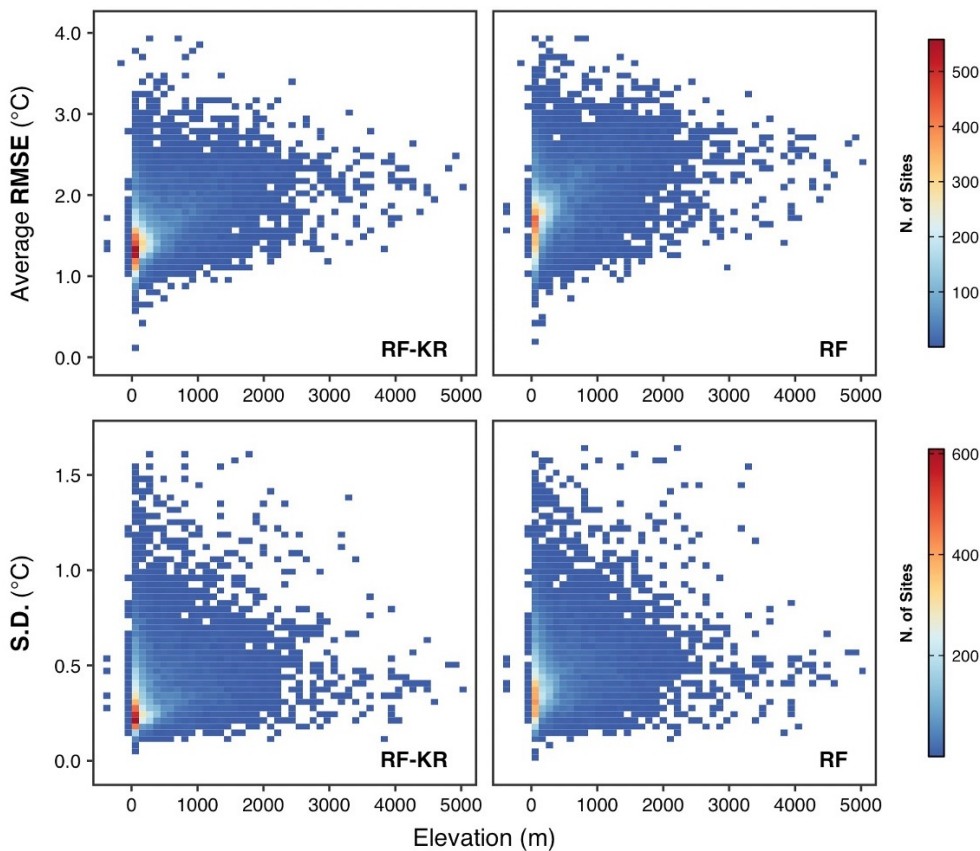

**Figure 8. The relationship between the site-level RMSE and elevations (top panels), and the relationship between the variability (S.D.) in the site-level RMSE and elevations (bottom panels) for the hybrid models used to generate GHRSAT and the RF models constructed in the first stage of the hybrid models.**

To quantitatively analyze the overall variability of the site-level RMSE for the sites across different elevations, the averaged RMSE and S.D for the site-level RMSE for different ranges of elevations are summarized in Table 2. The averaged RMSE



for the RF-KR models and the RF models increases with elevations from 1.55 °C to 2.14 °C, 1.82 °C to 2.40 °C, respectively. In different ranges of elevations, the RF-KR models consistently achieve higher predictive performance than the RF models,

indicating the significance of the hybrid modelling strategy for estimating SAT. The averaged RMSE reduced by the RF-KR models with respect to the RF models across different elevation ranges is between 0.22 °C to 0.30 °C, which corresponds to the 10%-15% marginal improvements to the predictive performance of estimating hourly SAT compared to the RF models. In contrast to the RF models, the RF-KR models also have low variability in the averaged RMSE for the site-level RMSE across different ranges of elevations.

**Table 2. The averaged site-level RMSE and the standard deviance (S.D.) of the site-level RMSE for the hybrid models used to generate GHRSAT and the RF models constructed in the first stage of the hybrid models for the sites within different ranges of eleveations.**

| Elevation Range | RMSE (°C) | | | S.D. (°C) | | |
|---|---|---|---|---|---|---|
| | RF-KR | RF | Dec. (%) | RF-KR | RF | Dec. (%) |
| < 1000 | 1.55 | 1.82 | 0.27 (15%) | 0.37 | 0.45 | 0.08 (18%) |
| 1000–2000 | 1.94 | 2.24 | 0.30 (13%) | 0.47 | 0.54 | 0.07 (13%) |
| 2000–3000 | 2.06 | 2.31 | 0.25 (11%) | 0.51 | 0.57 | 0.06 (11%) |
| 3000–4000 | 2.07 | 2.29 | 0.22 (10%) | 0.47 | 0.52 | 0.05 (10%) |
| > 4000 | 2.14 | 2.40 | 0.26 (11%) | 0.42 | 0.49 | 0.07 (14%) |

### 4.3 Impacts of learning parameters

Tuning the learning parameters or hyper-parameters for machine learning algorithms is a vital step involved in applying the algorithms in modelling tasks. It is worth noting that there is great variability in the modelling tasks due to the diversity of the environmental settings and scales for geographic areas. Thus, a machine learning model optimally developed using the parameters tuned using the samples for a specific area probably fails to have high generalization capability for other areas. In general, the models developed in a study achieving very satisfactory validation accuracy by massively tuning the parameters

for the models will have limited implications for other studies focusing on other areas with different data sources. In the field of remotely sensed estimation of SAT, studies have been performed to develop the estimation models for a variety of areas with different scales, such as global land areas (Hooker et al., 2018; Kilibarda et al., 2014), the Tibetan Plateau (Qin et al., 2023b), central north America (Zhang and Du, 2022a) and the Antarctic (Meyer et al., 2016; Nielsen et al., 2023). However, the estimation performance of the models developed in these studies should not examined only from the viewpoint of

statistical methods, and the scales and data sources of the studies need to be considered.

From the practical perspective of modelling using a machine learning algorithm, it is infeasible to exhaustively tune all learning parameters for the algorithm, especially for the case in which large numbers of models need to be developed using

different samples, and the sizes of the samples are huge. In this study, the hybrid models for generating the GHRSAT hourly
SAT estimates were developed across the months between 2011 and 2023 for each task region. Considering the insensitivity
of random forest to parameter tuning and model overfitting (Meyer et al., 2018; Meyer and Pebesma, 2022), the RF models
constructed in the first stage of the hybrid models for on region utilized the same set of learning parameters, which was
optimally tuned from a parameter grid of core parameters (Table S1) for January of 2020. As such, we cross-validated the
RF models for each task region using 72 different sets of learning parameters, and the hybrid models separately based on the
RF models were also validated. Fig. 9 shows the variability of the overall performance across the RF models and the RF-KR
models cross-validated using different sets of parameters.

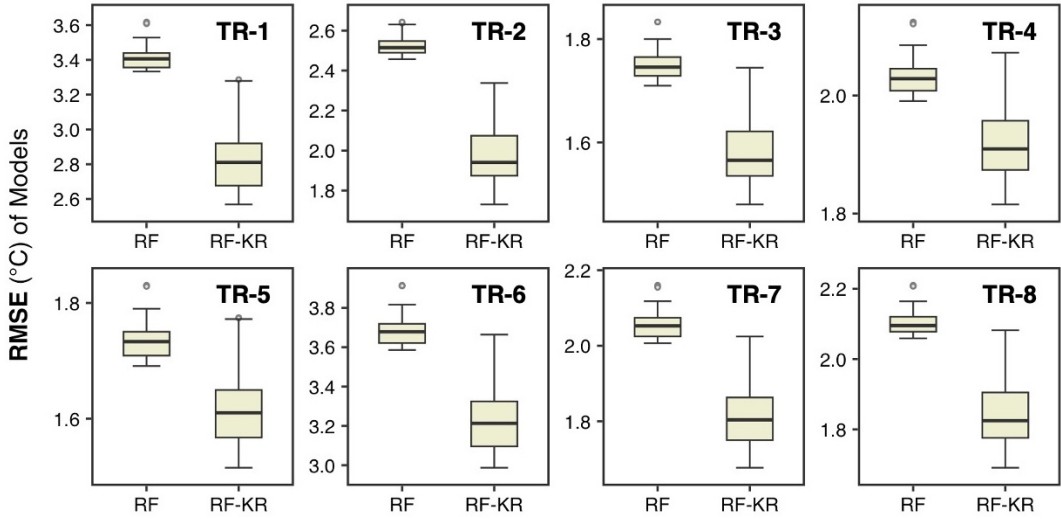

**Figure 9. Variability in the overall predictive performance of the hybrid models (RF-KR) cross-validated using the
samples for January of 2020 for each task region and the different sets of learning parameters for the RF models
developed in the first stage of RF-KR.**

The variability of the overall performance across the models for different regions shown in Fig. 9 as the same as Fig. 4 is
significant, and indicates that the predictive capability the estimation models for SAT based on statistical methods is severely
impacted by the characteristics of geographic areas and the samples extracted for model training. The average RMSE for the
RF models using different tuning parameters is respectively 3.41 °C and 3.68 °C for the regions of TR-1 and TR-6, which
are located in polar areas with very limited coverage of stations. The RF models for the other six regions have the average
RMSE of 1.74–2.52 °C. We confirmed the insensitivity of random forest to the tuning of hyper-parameters in the modelling
of  hourly SAT (Fig. 9), especially for regions with abundant samples from ground stations. The range of the overall RMSE
for the RF models using different tuning parameters is 0.28 °C for TR-1 and 0.33 °C for TR-6, while the range for the other
six regions is only between 0.12 °C and 0.19 °C (Table 3). We can substantially improve the predictive performance of
estimating SAT for the regions of TR-1 and TR-6 by properly tuning the core learning parameters for the RF models,
although relatively limited improvements to the performance can be achieved for other regions.



However, as shown in Fig. 9, the hybrid models (RF-KR) for each task region based on the RF models tuned using different parameters show remarkably larger variability in the predictive performance, compared to the RF models. The range of the RMSE for the RF models across different regions is 0.12–0.19 °C, while the range of the RMSE for the RF-KR models is between 0.26 and 0.72 °C. The range of RMSE for the RF models is more than twice than that for the RF-KR models for

each region. The RF-KR models for TR-1 and TR-6 have obviously higher variability than other regions with the RMSE ranging from 0.72 °C to 0.68 °C. The RMSE range for the RF-KR models is 0.61 °C for TR-2, which is more than three times the range for the RF models. Although there is great variability in the overall predictive performance of the hybrid models due to the tuning of different parameters for the RF models in the first stage, the hybrid models based on the RF models consistently achieve higher predictive performance compared to the RF models. In terms of the average RMSE for

the models for each region, the prediction errors in estimating hourly SAT are decreased by 0.11–0.58 °C with respect to the RF models when using the hybrid models, and the ratio of relative decrease in the average RMSE compared to the RF models is between 5.4% and 21.8%. The prediction errors in estimating hourly SAT for TR-1 and TR-6, for which the RF models performed with the average RMSE more than 3 °C, can even be respectively decreased by 0.58 °C and 0.45 °C in the averaged RMSE when using the hybrid models.

**Table 3. Statsitics of the variability in predictive performance for the hybrid models shown in Fig. 9.**

| Task Region | RF | | RF-KR | | Mean Decrease | Decrease Ratio (%) |
|---|---|---|---|---|---|---|
| | Mean | Range | Mean | Range | | |
| TR-1 | 3.41 | 0.28 | 2.83 | 0.72 | 0.58 | 17.0% |
| TR-2 | 2.52 | 0.19 | 1.97 | 0.61 | 0.55 | 21.8% |
| TR-3 | 1.75 | 0.12 | 1.58 | 0.26 | 0.17 | 9.7% |
| TR-4 | 2.03 | 0.13 | 1.92 | 0.26 | 0.11 | 5.4% |
| TR-5 | 1.74 | 0.14 | 1.61 | 0.26 | 0.13 | 7.5% |
| TR-6 | 3.68 | 0.33 | 3.23 | 0.68 | 0.45 | 12.2% |
| TR-7 | 2.06 | 0.15 | 1.81 | 0.35 | 0.25 | 12.1% |
| TR-8 | 2.10 | 0.15 | 1.84 | 0.39 | 0.26 | 12.4% |

We analyzed the relationship of the overall performance between the RF-KR models and the RF models constructed in the first stage of RF-KR using different sets of learning parameters. As shown in Fig. 10, we see that there is the approximately linear relationship of the performance between the RF models and the RF-KR models for each task region, suggesting that

the predictive performance of the hybrid models can be gradually improved by decreasing the prediction errors in the RF models. Furthermore, one unit of reduction in the RMSE for the RF models can result in more than one unit of reduction in the RMSE for the RF-KR models, which can be seen from the slope of the linear relationship shown in Fig. 10. For example, the two RF models tuned with the lowest and highest predictive performance for TR-2, which are labelled as L and H in Fig.





10, have the RMSE of 2.64 °C and 2.45 °C, respectively. The two RF-KR models separately based on the two RF models
achieve the RMSE of 2.34 °C and 1.73 °C. In such case, improving the performance of the RF models for TR-2 by 0.19 °C
with the RMSE decreasing from 2.64 °C to 2.45 °C results in the significant improvement to the RF-KR models by 0.72 °C
with the RMSE decreasing from 2.34 °C to 1.73 °C. Therefore, the two-stage hybrid modelling strategy for developing the
models to generate the GHRSAT product can significantly improve the predictive performance of estimating SAT when the
models developed in the first stage are tuned with lower errors.

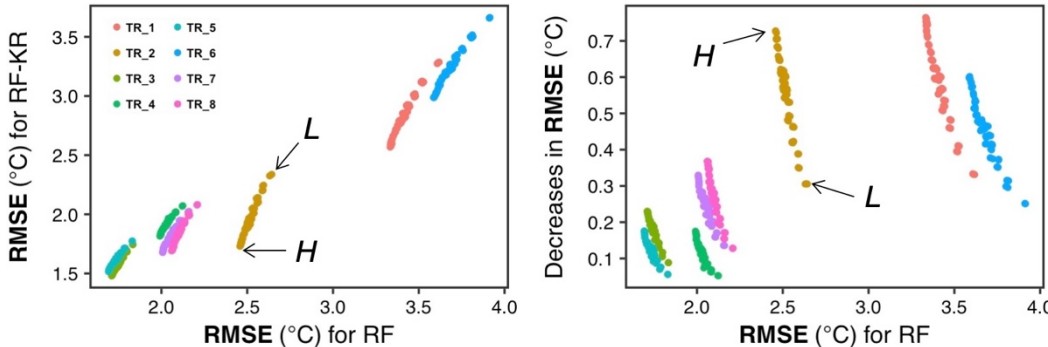


**Figure 10. The relationship of the overall predictive performance between the RF models and the RF-KR hybrid
models based on the RF models tuned using different learning parameters (left). The relationship between the
overall predictive performance of the RF models and the improvements to the performance (decrease in RMSE)
attributed to the hybrid models. The two points labelled with L and H represent the RF models for TR-2 with the**
**lowest and highest performance, respectively.**

## 4.4 Kriging for large spatial data

Kriging is known as the optimum interpolation technique, and involves inversion of variance-covariance matrices, which is
very computationally expensive for kriging large numbers of point samples. We reconstructed the global hourly estimates of
SAT in the GHRSAT product by developing two types of hybrid models including RF-OK and RF-FRK, which are designed
for the task regions with limited stations and the regions with large numbers of stations, respectively. The RF-FRK models
were designed with the aim of efficiently modelling the station residuals in the first-stage RF models using the FRK method
for each hour in the period of 2011–2023. Although FRK has the merit of modelling spatial non-stationarity, FRK relies on
the simplification of kriging equations to significantly reduce the computational cost of solving the equations. Therefore, it is
highly likely that compared to the conventional ordinary kriging, applying FRK to the modelling of station residuals in the
second stage of the hybrid models will result in the reduction in the predictive performance of estimating SAT for the hybrid
models. To explore the difference in the performance for the hybrid models caused by the application of the kriging method
of OK or FRK, we developed and cross-validated both the RF-OK models and RF-FRK models for the regions TR-2, TR-4
and TR-7 across the months of the year 2020. The models of RF-OK and RF-FRK were based on the same RF models for
each task region and each month.

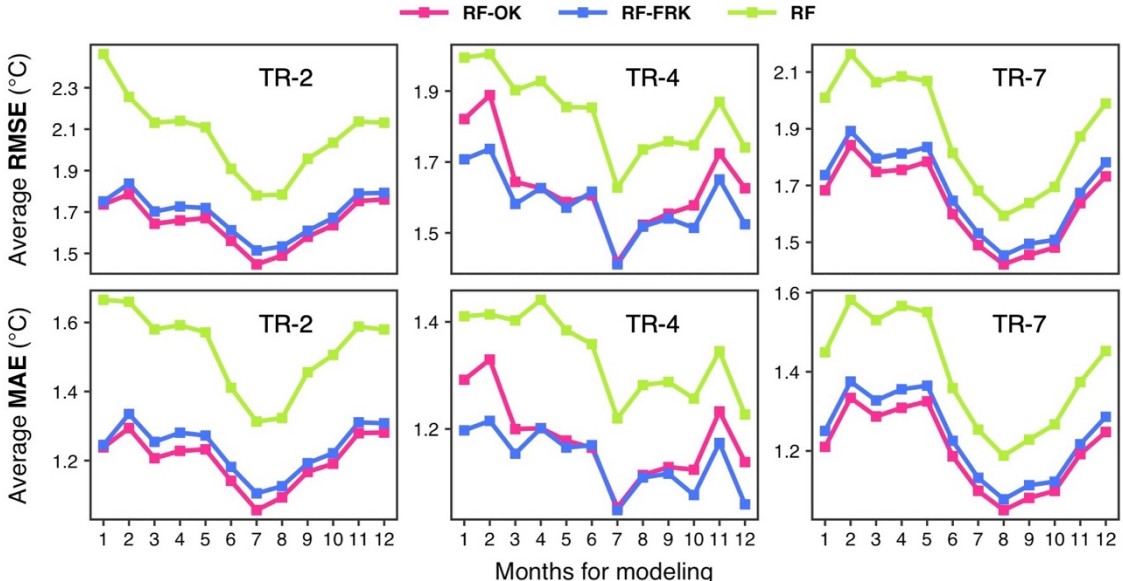


**Figure 11. Comparison of ordinary kriging (OK) and fixed rank kriging (FRK) in developing the hybrid models for the task regions with large numbers of stations including TR-2, TR-4 and TR-7 across the months in 2020. The hybrid models (RF-OK and RF-FRK) were based on the same RF models constructed in the first stage of the hybrid models.**

As shown in Fig. 11, there are slight differences in the overall predictive performance between the RF-OK models and the RF-FRK models across the months in 2020 for each region. The RF-OK models across the months have the overall RMSE of 1.45–1.79 °C, 1.42–1.89 °C, and 1.42–1.84 °C for TR-2, TR-4, and TR-7, respectively. In contrast, the RF-FRK models across the months achieve the overall RMSE of 1.51–1.84 °C, 1.41–1.74 °C, and 1.45–1.89 °C for TR-2, TR-4, and TR-7, respectively. The average difference between the RF-OK models and the RF-FRK models for the three regions in terms of

RMSE and MAE is only about 0.04–0.05 °C, and 0.04–0.06 °C, respectively. Both the RF-OK models and the RF-FRK hybrid models perform significantly better than the RF models constructed in the first stage of the hybrid models. The RF models across the months have the average RMSE of 2.07 °C, 1.83 °C and 1.89 °C for TR-2, TR-4, and TR-7, respectively. The hybrid models of RF-KR and RF-FRK, which further modelled the station residuals in the RF models, reduce the errors of estimating hourly SAT in terms of average RMSE by about 0.38–0.43 °C, 0.2–0.25 °C and 0.21–0.25 °C for TR-2, TR-4,

and TR-7, respectively.

Due to the simplification of kriging equations by FRK, it is assumed that the hybrid models using FRK will perform poorer than the hybrid models using OK for the modelling of station residuals. Fig. 11 indicates that the RF-OK models consistently perform better than the RF-FRK models across the months for TR-2 and TR-7 with the reduction in the average RMSE by 0.05 °C and 0.04 °C, respectively. However, the RF-FRK models generally achieve lower prediction errors than the RF-OK

models across the months for TR-4 with the average RMSE reduced by 0.05 °C. In particular, there is apparent difference in the predictive performance between the RF-OK models and the RF-FRK models for TR-4 across the winter months, and the

difference in RMSE ranges from about 0 °C to 0.15 °C. In addition to the performance of the RF models, the performance of the hybrid models is also related to the modelling of the station residuals in the RF models. As the spatial structures in hourly SAT have been largely modelled by RF models, the signal of the structures in the station residuals is usually weak, and the intrinsic noises (nugget effects) associated with the station residuals will be a major part of the residuals. As such, the station residuals cannot be well modelled in some circumstances by ordinary kriging, which utilize one global variogram model to represent the spatial auto-correlation between two locations (Oliver and Webster, 2014). The FRK method represents the spatial structures of auto-correlation by several spatial basis functions at different scales (Zammit-Mangion et al., 2018), which may result in better modelling of the residuals with apparent structures at different scales and non-stationarity.

**Figure 12. Spatial distribution of the predictive performance at the site-level for the RF-OK and RF-FRK hybrid models for different task regions, which were based on the same RF models developed across the months in 2020. The RMSE for the sites were spatially binned into 1-degree grid boxes for better visualization.**

The difference in the predictive performance between the RF-OK models and the RF-FRK models for each region is further
assess at the level of sites and the daily scale. The cross-validated samples for the models across the months in 2020 were
averaged to compute the RMSE for each site (Fig. 12) and for each day (Fig. 13). Fig. 12 shows the distribution of the
averaged RMSE by spatially binning sites into 1-degree grid boxes for better visualization. We see that the difference in the
spatial distribution of the averaged RMSE at the site-level between RF-OK and RF-FRK is very slight for the three task
regions. The models of RF, RF-OK and RF-FRK for each region have very similar spatial pattern of prediction errors across
sites or the region. All models for TR-2 exhibit relatively higher prediction errors in the central areas of the western US, and
the models for TR-4 perform poorer in the southern areas of Europe and Norway with the site-level RMSE above 2 °C. The
models for TR-7 show very high prediction errors with the site-level RMSE of about 2–4.5 °C primarily in the areas of the
Pamir Plateau, the Tibetan Plateau, and the Mongolian Plateau. The areas with high prediction errors in the models for the
three regions are characterized by complex topography and atmospheric environments, and these areas are also subject to the
scarcity of ground stations. However, it can be observed from Fig. 12 that, compared to the RF models, the high prediction
errors in these areas are substantially reduced by the RF-OK or RF-FRK hybrid models.

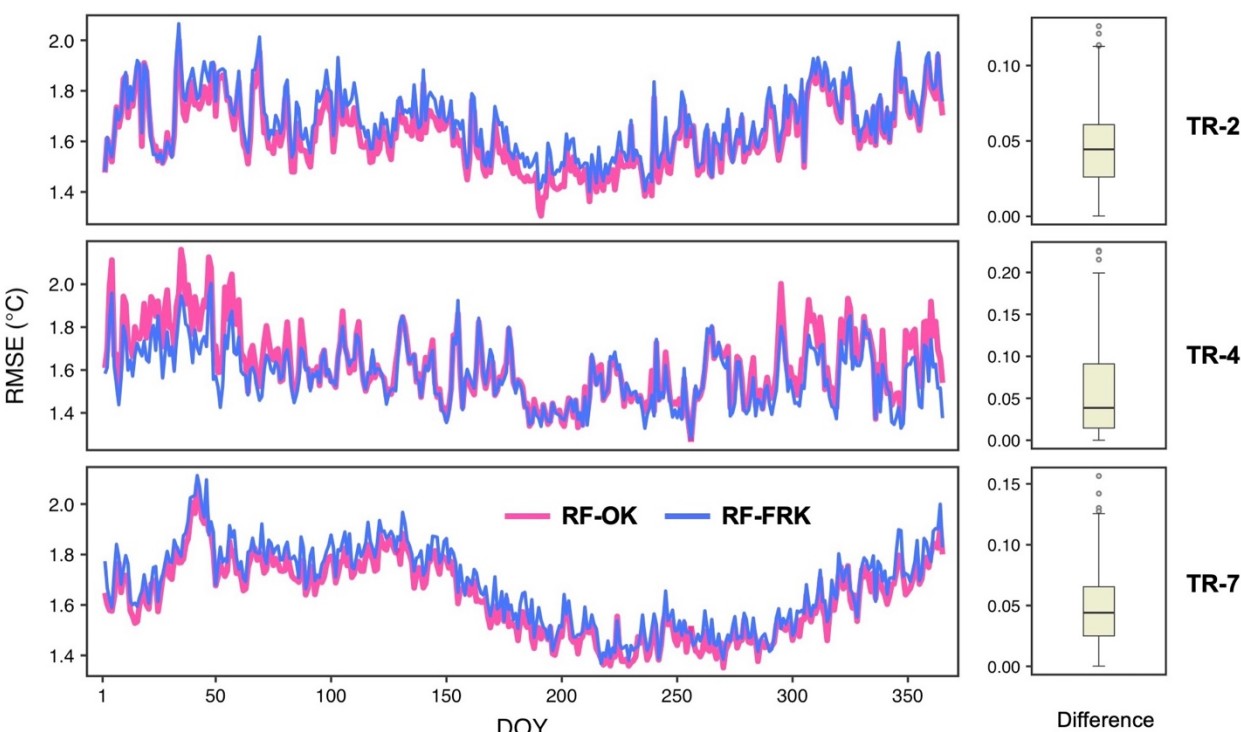

**Figure 13. Comparison of the predictive performance on the daily basis between the RF-OK and RF-FRK hybrid
models based on the same RF models for different task regions. The panels of boxplots in the right summarize the
overall distribution of the daily differences in the predictive performance between RF-OK and RF-FRK.**



As shown in Fig. 13, the RF-OK models exhibit very similar predictive performance with the RF-FRK models across the days in 2020 for the three task regions. However, there still exist slight differences in the predictive performance between RF-FRK and RF-OK. For the hybrid models for TR-2, the RK-OK models are better than the RF-FRK models for the days around the DOY of 200. For TR-4, the RF-OK models perform slightly poorer than the RF-FRK models for the days around the DOY of 50 and 350. For TR-7, the RF-OK models are slightly better than the RF-RFK models across all days of the year 2020. The boxplots Fig. 13 show the distribution of the differences in the daily RMSE between RF-OK and RF-FRK for the three task regions. The average of the differences in daily RMSE for each region is below 0.05 °C, and the differences are primarily below 0.1 °C for the days in 2020.

The previous results demonstrate the superiority and generality of developing the estimation models by the hybrid modelling strategy for estimating SAT for both small-scale areas with limited coverage of ground stations and the large-scale areas with large numbers of stations. The hybrid models that integrate machine learning methods and the kriging modelling of residuals at stations can substantially improve the predictive performance of estimating SAT compared to the models only based on statistical methods. For the case of modelling SAT in the areas with large numbers of stations, computationally efficient kriging techniques such as FRK (Cressie and Johannesson, 2008) and NNGP (Datta et al., 2016) can be employed to develop hybrid models.

## 5 Data availability

The GHRSAT product generated by this study contains all-sky hourly estimates of SAT between 2011 and 2023 for the global land areas excluding Antarctica, and is available at https://doi.org/10.11888/RemoteSen.tpdc.301540 (Zhang, 2024). Data of the global SAT estimates for the 24 hours in each day were organized into a data file in the format of netCDF. The data files for one month were compressed into a zip file. The total size of all compressed zip files for the GHRSAT product is about 1.2 T.

## 6 Conclusions

This study for the first time generated the GHRSAT product of all-sky hourly estimates of SAT between 2011 and 2023 for the global land areas. The hourly estimates were reconstructed using the estimation models designed by the hybrid modelling strategy that integrates random forest and two types of kriging techniques. The hybrid models designed by the strategy were constructed in two stages, and by integrating the training of a random forest model (RF) in the first stage using the samples extracted at ground stations with the kriging modelling of the station residuals in the first-stage model. The global land areas excluding Antarctica were divided into 8 task regions, and considering the scale and data volume of the modelling in this study, we developed two types of hybrid models including RF-OK and RF-FRK on the monthly basis for different task regions across the months between 2011 and 2023. RF-OK and RF-FRK perform the modelling of the station residuals in the



RF models using ordinary kriging for the regions with limited stations and fixed rank kriging for the regions with large numbers of stations, respectively. All estimation models including the hybrid models and the RF models constructed in the first stage were fully cross-validated for assessing the predictive performance of the models in estimating hourly SAT. We analyzed the impacts of tuning the learning parameters for the first-stage RF models and the influence of modelling the station residuals in the RF models by different kriging techniques on the predictive performance of the hybrid models. We expect the GHRSAT product generated in this study will be applied in various fields such as environmental assessments and hydrological studies. The main results of this study are concluded as follows.

The hybrid models for generating GHRSAT achieve the predictive performance for different regions with the overall RMSE between 1.48 °C to 2.28 °C. Our results demonstrate that the hybrid modelling strategy can greatly improve the predictive performance for estimating SAT compared to the modelling of SAT only based on statistical methods. In average, the hybrid models improve the performance for estimating global hourly SAT by 0.18–0.41 °C in terms of RMSE with respect to the RF models constructed in the first stage of the hybrid models. There is the temporal variability in the performance across the months between 2011 and 2023 and the spatial variability in the performance across the ground stations in different areas. However, compared to the RF models, the spatio-temporal variability is significantly reduced by the hybrid models.

We found that although tuning the parameters for the first-stage RF models has limited impacts on the performance of the models, it can drastically influence the performance of the hybrid models. There is the approximately linear relationship of the performance between the RF models and the hybrid models developed for each region, and the performance of the hybrid models can be greatly improved when the RF models developed in the first stage of the hybrid models slightly tuned with lower errors. In addition, we developed the RF-FRK models using the computationally efficient FRK method with the aim of modelling the station residuals in the RF models for regions with large numbers of stations. FRK reduce the computational cost at the sacrifice of simplifying kriging equations, which may result in the reduction in the performance of the hybrid models. However, we found that there are slight differences in the predictive performance between the RF-OK models and the RF-FRK models, which are based on the same RF models, and the average difference in terms of RMSE is only about 0.04–0.05 °C. The previous results demonstrate the superiority and generality of developing the estimation models by the hybrid modelling strategy for estimating SAT for both small-scale areas with limited coverage of stations and the large-scale areas with large numbers of stations.

## Acknowledgements

We are grateful to the study by Jia et al. to generate the GHA-LST dataset, which is the basis of the hybrid estimation models developed in this study for reconstructing the GHRSAT hourly estimates of surface air temperature. We thank NOAA NCEI (National Centers for Environmental Information) for developing and maintaining the ISD dataset that contains quality-controlled hourly observations of air temperature at the ground stations across the global land areas.



## Author contributions

ZZ conceptualized and designed the overall research framework of this study. ZZ developed the hybrid estimation models. A
systematic assessment the estimation models were performed by ZZ, KW and Zihan Y. Formal data analysis, interpretation,
and visualization were primarily performed by ZZ and CL with the support from Zishang Y. ZZ and KW applied the
estimation models for generating the GHRSAT dataset. The dataset was organized and archived by ZZ. ZZ wrote the paper
with contributions from all of the co-authors.

## Competing interests

The contact author has declared that none of the authors has any competing interests.

## Financial support

This work was supported by the Natural Science Research of Jiangsu Higher Education Institutions of China (grant number
23KJB420004) and the Startup Foundation for Introducing Talent of NUIST (grant number 2022r040).

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
