# Peer review of "GHRSAT: the first global hourly dataset of all-sky remotely sensed estimates of surface air temperature"

_Earth System Science Data, 2024_

## Referee Comment (RC1)

The manuscript "GHRSAT: the first global hourly dataset of all-sky remotely sensed estimates of surface air temperature" developed a hybrid method integrate random forest models and kriging techniques to estimate all-weather air temperature, and generate global all-weather air temperature products from 2011-2023. The proposed method improved the model accuracy compared to traditional RF algorithom. The content structure of the preprint is clear, the method is innovative and the topic is meaningful. My comments and questions for clarification can be found below.

**Comments:**
1. Line 120: The author uses zoning modeling and mentions that the zoning basis also refers to station density. I am confused that the model building effect depends largely on the representativeness of the sample. Is it reasonable to use low-density sites for modeling? In addition, will zoning modeling lead to boundary effects between regions? Why not build a global unified model?
2. Line 158: Air temperature is related to many factors. The variables input into the model in this article are only LST, NDVI, latitude and longitude, elevation and hour of a day. What is the basis for selecting these variables? Among them, only LST and hour of a day change over hours. Is the result mostly dependent on LST? Please show the feature importance of the models.
3. Line 255: In the validation part, the samples were randomly divided into ten parts, one of which was used to validate the model, which means that the training samples may include all sites, and there is no completely independent site for validation. What is the prediction accuracy of this method in non-site areas?
4. Line 289: The author developed 156 models for each region. The temporal variation of air temperature has certain regularities, and data from the same period in different years may provide effective information. Why does the author establish a separate model for each month in 2011-2023?
5. Section 4.2: The spatial validation in the preprint is based on the station scale, which cannot reflect the continuity of the generated product and the estimated effect of non-site areas. Please further prove it at the spatial scale.
6. The preprint lacks data cross-validation. For example, the air temperature estimated from geostationary satellites or reanalysis data all have hourly air temperatures. Please compare the with the published air temperature data or methods.

---

## Author Comment (AC1)

**Responses to RC1**

*Dear Reviewer #1:*

*Thanks very much for your time on reviewing our manuscript. We sincerely thank the reviewer for your efforts on the reviewing of our manuscript. We deeply appreciate your valuable comments on our manuscript, and we have carefully revised the manuscript according to the comments. The point-by-point responses to your comments are provided in this document.*

*Best regards,*
*Zhenwei Zhang*
*Nanjing University of Information Science & Technology*

The manuscript "GHRSAT: the first global hourly dataset of all-sky remotely sensed estimates of surface air temperature" developed a hybrid method integrate random forest models and kriging techniques to estimate all-weather air temperature, and generate global all-weather air temperature products from 2011-2023. The proposed method improved the model accuracy compared to traditional RF algorithm. The content structure of the preprint is clear, the method is innovative, and the topic is meaningful. My comments and questions for clarification can be found below.

Comments:

**1.** Line 120: The author uses zoning modeling and mentions that the zoning basis also refers to station density. I am confused that the model building effect depends largely on the representativeness of the sample. Is it reasonable to use low-density sites for modeling? In addition, will zoning modeling lead to boundary effects between regions? Why not build a global unified model ?

**Responses #1**: Thanks for your valuable comments. As you mentioned, representativeness of samples severely influences model building processes. Additionally, quantity and quality of the samples also impact model building. As stated in Lines 190-191 of our original manuscript, we obtained about 0.9 billion matched samples at ground stations for the global land study areas in the 2011–2023 period, and the average number of samples for each calendar month is more than 5.9 million. **It is computationally infeasible to build a global unified model using all matched samples.** Given the large numbers of samples across the global land areas in the long-term period, we chose to develop estimation models separately for eight different task regions **not only considering the computation issue, but also based on the following concerns**: locally developed models have high adaptability to specific local land areas. As an analogy, the reason why GWR (geographically weighted regression) has been widely adopted

for various modeling tasks in the broad fields of geosciences is that GWR exhibits high modeling capacity than global linear regression models. In essence, the GWR method is implemented by building a linear model for each sample point using the nearby samples within the local neighborhood of the point. The study for estimating long-term SAT in China (Yao R. et al., 2020) conducted comparative experiments indicating the advantage of locally modelling strategies in terms of prediction performance. For the modeling tasks involved in our study, as the global land areas cover a wide variety of geographical settings and backgrounds, a set of locally developed models targeted for each specific region will have high adaptability. **Therefore, it could be a good choice to adopt a locally (both time and geographic space) modeling strategy when building models for large-scale areas with huge number of samples.** Similarly, the study by Zhang T. et al. (2022) on estimating daily SAT for global land areas also developed models separately for different areas.

In fact, the locally or zoning modeling strategy try to build region-specific estimating models with high adaptability to each region. The boundary effects of the modelling strategy are inevitable as all modelling tasks involved in not only our work, but also other research fields have to be targeted for a predefined or manually selected study area.

We really appreciate your concern about the reasonability of modeling for regions with low-density sites. As you mentioned, the samples from low-density sites will severely impact the sampling representativeness. Generally, models developed using samples from study areas with low-density sites will have large errors and estimation uncertainties (see the reported performance metrics for regions of TR-1 and TR-6 in Figure 4), which is primarily due to severely inadequate sampling representativeness. Despite the issue for the areas with limited sites, such as polar areas and high-altitude areas, it is worth exploring to develop models for these regions, although the models have large errors and uncertainties. There are studies have been conducted for building SAT models for the areas with very limited stations (Nielsen E. B., et al, 2023; Meyer H., et al, 2016). **We admit that at present, building models for these areas with satisfactory performance is very challenging**, considering the limited stations installed in these areas. With more autonomous ground stations installed in the harsh polar and high-altitude areas, the estimation of SAT for these areas will be gradually improved. **Lastly, high-quality observations from ground stations are the basis for various research fields, and the scarcity of ground stations is the common issue for these fields**. The studies conducted for these fields can only be based on the available ground sites. For example, the studies on the estimation of surface radiation for large-scale areas all depend on very limited number of ground flux sites. Recently, the study on estimating surface long-wave downward radiation (Zeng Q. et al, 2024) published in the ISPRS journal utilized all available 51 ground flux sites across the central Asia. Thus, to advance the studies in the broad geoscientific community, it is very important for governments and research agencies to further improve ground observation networks, especially for polar and high-altitude areas.

Finally, we thank the valuable comments again. We have stated the reasons for adopting the locally model building strategy (Lines 122-133) and discussed the issue of sampling representativeness for building SAT estimation models for regions with limited stations (Lines 360-371).

*Yao, R., Wang, L., Huang, X., Li, L., Sun, J., Wu, X., and Jiang, W.: Developing a temporally accurate air temperature dataset for Mainland China, Sci. Total Environ., 706, 136037, https://doi.org/10.1016/j.scitotenv.2019.136037, 2020.*

*Zhang, T., Zhou, Y., Zhao, K., Zhu, Z., Chen, G., Hu, J., and Wang, L.: A global dataset of daily maximum and minimum near-surface air temperature at 1 km resolution over land (2003–2020), Earth Syst. Sci. Data, 14, 5637–5649, https://doi.org/10.5194/essd-14-5637-2022, 2022.*

*Nielsen, E. B., Katurji, M., Zawar-Reza, P., and Meyer, H.: Antarctic daily mesoscale air temperature dataset derived from MODIS land and ice surface temperature, Sci Data, 10, 833, https://doi.org/10.1038/s41597-023-02720-z, 2023.*

*Meyer, H., Katurji, M., Appelhans, T., Müller, M., Nauss, T., Roudier, P., and Zawar-Reza, P.: Mapping Daily Air Temperature for Antarctica Based on MODIS LST, Remote Sens., 8, 732, https://doi.org/10.3390/rs8090732, 2016.*

*Zeng, Q., Cheng, J., Sun, H., and Dong, S.: An integrated framework for estimating the hourly all-time cloudy-sky surface long-wave downward radiation for Fengyun-4A/AGRI, Remote Sensing of Environment, 312, 114319, https://doi.org/10.1016/j.rse.2024.114319, 2024.*

**2.** Line 158: Air temperature is related to many factors. The variables input into the model in this article are only LST, NDVI, latitude and longitude, elevation and hour of a day. What is the basis for selecting these variables? Among them, only LST and hour of a day change over hours. Is the result mostly dependent on LST? Please show the feature importance of the models.

**Responses #2**: Thanks for your valuable comments. The primary fundamental of selecting the input variables for modeling SAT is based on their connection to SAT, and specifically considers whether incorporating the variables into SAT estimation models will contribute the predictive performance of the models. As our study aimed at building estimation models for global land areas, it is inevitable to only consider the variables for which datasets are available at the global scale in the time period 2011-2023.

There are some differences in the selection of variables for SAT estimation among previous studies, which **is primary due to the localized consideration of modeling SAT for specific study areas and the constraints of data availabilit**y. For examples, previous studies have developed SAT estimation models considering variables for satellite-based snow cover (Wang W. et al., 2025) and surface structural properties derived from lidar data (Venter Z. S., et al., 2020). However, the models utilizing these variables are only restricted to the study areas that the studies focused on, and cannot be generalized to other regions due to data unavailability for these variables in other regions.

The auxiliary variables used in our study have been widely used in previous studies for building SAT estimation models. More importantly, data for the auxiliary variables used in our study are available at the global scale, and can be easily and publicly accessed online. The

hourly LST is the core input for our SAT estimation models, and as you pointed out, these models are primarily dependent on LST. **We have provided a figure for illustrating the feature importance of the input variables for the SAT models** (see Figure S3 in the revised supplement file), and we have rewritten some parts of Sec. 2.3 to more clearly state our consideration for selecting the auxiliary variables for modeling hourly SAT in our study (see Lines 155-166).

With more earth observation satellites operating at high temporal revisit cycles for large-scale areas in the future, we expect that the hourly estimation of SAT will be significantly improved by more available high-temporal satellite-based data for land surface properties relevant for modeling SAT.

Wang, W., Brönnimann, S., Zhou, J., Li, S., and Wang, Z.: Near-surface air temperature estimation for areas with sparse observations based on transfer learning, ISPRS Journal of Photogrammetry and Remote Sensing, 220, 712–727, https://doi.org/10.1016/j.isprsjprs.2025.01.021, 2025.

Venter, Z. S., Brousse, O., Esau, I., and Meier, F.: Hyperlocal mapping of urban air temperature using remote sensing and crowdsourced weather data, Remote Sens. Environ., 242, 111791, https://doi.org/10.1016/j.rse.2020.111791, 2020.

**3**. Line 255:In the validation part, the samples were randomly divided into ten parts, one of which was used to validate the model, which means that the training samples may include all sites, and there is no completely independent site for validation. What is the prediction accuracy of this method in non-site areas?

**Responses #3**: Thanks for your important questions. According to your comments, we have **additionally performed site-based cross-validation (CV) for all models developed in our study**. In the site-based cross-validation, the sites were first randomly divided into ten sets, and samples from the sites in each set are treated as one fold of samples (see Lines 289-294, in Sec. 3.3). For site-based CV, models are validated by completely independent sites. Overall, the averaged performance of the hybrid models for different regions ranges from 1.87 °C to 2.62 °C under site-based CV. The validation results for sited-based CV have been discussed in our revised manuscript (Lines 26-29 in the Abstract, Lines 332-359 in Sec. 4.1, and Lines 694-699 in Sec. 6). Fig. 4 has been revised to contain the overall validation results for our models under both sample-based and site-based cross-validation (Lines 380-385), and some additional figures have been added in our revised supplement file (Fig. S4, S5, S6).

**4.** Line 289: The author developed 156 models for each region. The temporal variation of air temperature has certain regularities, and data from the same period in different years may provide effective information. Why does the author establish a separate model for each month in 2011-2023?

**Responses #4**: Thanks for your important comments. The comments are related to the locally

(both time and geographical areas) modelling strategy adopted in our study, which has been discussed in our Responses #1. We obtained about 0.9 billion matched samples at ground stations for the global land study areas in the 2011–2023 period. In addition to the consideration of computational efficiency for model training, developing models built for each month will have high adaptability to the month.

**5.** Section 4.2: The spatial validation in the preprint is based on the station scale, which cannot reflect the continuity of the generated product and the estimated effect of non-site areas. Please further prove it at the spatial scale.

**Responses #5**: Thanks for your valuable comments. We really appreciate the comments. SAT is estimated under the assumption that the model fitted using in situ samples generalizes to other areas (pixels) without ground samples, which is the basis for all studies in the field of NSAT estimation. It is hard to reflect the continuity of the generated product in term of estimation errors. In other words, as there are no abundant and very high-density ground stations available at the global scale, it is the rare case that each ground pixel contains several ground stations. Therefore, the estimation errors for non-site areas can only be represented by the cross-validation results for the SAT models. We do think that spatially quantification of estimation errors across the areas is important. The cross-validation results for SAT models can be analyzed at the site-level to indirectly exhibit how the models will perform across different areas. For example, see Fig. 7 from Kilibarda et al. (2014) and Fig. 6 from Yao et al. (2023).

*Kilibarda, M., Hengl, T., Heuvelink, G. B. M., Gräler, B., Pebesma, E., Perčec Tadić, M., and Bajat, B.: Spatio-temporal interpolation of daily temperatures for global land areas at 1 km resolution, J. Geophys. Res.: Atmos., 119, 2294–2313, https://doi.org/10.1002/2013JD020803, 2014.*

*Yao, R., Wang, L., Huang, X., Cao, Q., Wei, J., He, P., Wang, S., and Wang, L.: Global seamless and high-resolution temperature dataset (GSHTD), 2001–2020, Remote Sens. Environ., 286, 113422, https://doi.org/10.1016/j.rse.2022.113422, 2023.*

**6.** The preprint lacks data cross-validation. For example, the air temperature estimated from geostationary satellites or reanalysis data all have hourly air temperatures. Please compare the with the published air temperature data or methods.

**Responses #6**: We very appreciate your comments. To offer a further validation of our models, we have additionally performed site-based cross-validation (CV) for all models developed in our study. For site-based CV, the models are assessed by completely independent sites. The validation results for sited-based CV have been discussed in our revised manuscript (Lines 26-29 in the Abstract, Lines 332-359 in Sec. 4.1, Lines 380-385, and Lines 694-699 in Sec. 6).

We agree with you that the comparison of our dataset with other SAT datasets (reanalysis or SAT from geostationary) is very important. However, there are no common, independent and

adequate station data held out for validating the estimated SAT developed by different researchers using different methodological approaches. That is, if we hold out some parts of ground stations for validating our estimated SAT data, **the validation of the estimated SAT data developed by previous studies using the same hold-out stations will not be objective and independent**, because it is unknow if the hold-out stations had been used to train the models for the estimated SAT data in the previous studies.

Reanalysis data have high temporal resolutions, but have relatively coarse spatial resolutions. More importantly, reanalysis data (such as ERA5, GLDAS) are generated by numerical models with assimilation of various observational data sources, such as ground-based meteorological observation, satellite data and sounding data from radiosondes. The organizations (for example, ECMWF, NOAA, NASA GMAO) for developing reanalysis data and corresponding assimilation systems have not disclosed the specific information on the ground station data assimilated into their systems. Thus, validation of reanalysis using ground stations **will not be independent, and has the risk of over-estimation of the accuracy of reanalysis data**. Although it is unknown as to the specific information on the assimilated ground station by reanalysis data, **it could be inferred that publicly available observational datasets for ground stations are very likely be assimilated, because it is easy and free to access these public observational ground datasets**.

---

## Author Comment (AC2)

**Responses to RC2**

*Dear Reviewer #2:*

*Thanks very much for your time on reviewing our manuscript. We sincerely thank the reviewer for your efforts on the reviewing of our manuscript. We deeply appreciate your valuable comments on our manuscript, and we have carefully revised the manuscript according to the comments. The point-by-point responses to your comments are provided in this document.*

*Best regards,*
*Zhenwei Zhang*
*Nanjing University of Information Science & Technology*

The manuscript essd-2024-548 has been reviewed. It presents the first global hourly surface air temperature dataset (GHRSAT) from 2011 to 2023. The manuscript demonstrates clear organization, rigorous logical flow, and natural transitions between sections. While the overall quality of the work is commendable, several critical issues require attention, as outlined in the following comments:

**Comments #1.** The core method of this manuscript is the RF-KR method, that is, the RF model is used to build the site-scale SAT estimation model, and then the residual is interpolated by the Kriging interpolation method to obtain pixel-by-pixel residual data. Logically, it is possible, but the validation is not sufficient, that is, whether the cross-validation used in the test process can be validated with independent data to explain its accuracy better.

**Responses #1**: Thanks for your valuable comments. To offer a further validation of our models, we have additionally performed site-based cross-validation (CV) for all models developed in our study. In the site-based cross-validation, the sites were first randomly divided into ten sets, and samples from the sites in each set are treated as one fold of samples (see Lines 289-294 in Sec. 3.3). For site-based CV, models are trained using samples from nine sets of ground station, and the models are then validated by samples from one remaining set of stations. Thus, the models are assessed by completely independent sites. The validation results for sited-based CV have been discussed in our revised manuscript (Lines 26-29 in the Abstract, Lines 332-359 in Sec. 4.1, and Lines 694-699 in Sec. 6). Fig. 4 has been revised to contain the overall validation results for our models under both sample-based and site-based cross-validation (Lines 384-389), and some additional figures have been added in our revised supplement file (Fig. S4, S5, S6).

**Comments #2.** In the process of model construction, the GHA-LST dataset is used as the main input, but it is recommended to discuss how its uncertainty will affect SAT estimation.

**Responses #2**: Thanks for valuable comments. We agree with you that uncertainty associated with model inputs, especially the GHA-LST reconstructed dataset, is important for the SAT models based on GHR-LST. But there is no available uncertainty information at the pixel level for GHR-LST, it is hard to design modeling experiments to quantitively analyze the impacts of GHR-LST reconstruction uncertainty on SAT estimation models. The SAT models use the ground measurements of SAT as the target variable, which is measured at high accuracy. In general, the errors and uncertainty of reconstructed LST and other inputs will be reflected by the predictive performance of the SAT models. We have briefly discussed this issue in our revised manuscript (Lines 411-413).

**Comments #3.** How to consider the spatial representativeness of the air temperature observed at the station on the 5km scale.

**Responses #3**: Thanks for pointing out this important question. The scale-mismatch between ground point-level station measurements and areal (pixel-level) observations from satellites is an innate and challenging issue not only for SAT estimation studies, but for a wide range of research fields using remote sensing data. SAT estimation models are trained using matched samples from ground stations. The scale-mismatch issue will **impact the spatial representativeness of the matched samples (or sampling errors), which further influences the predictive performance (errors) of the SAT models**. Better predictive performance for estimating SAT can be achieved by building models with more representative samples processed from high-density ground stations. However, it is very difficult to completely resolve the issue regarding spatial representativeness for ground sampling, given current status and developments of ground-based observational networks. To obtain more spatially representative samples for modeling SAT at the 5-km scale, very-high density of ground stations should be available for a study area. For example, the HiWATER research program (Li et al. 2013) established one network of high-density ground observation sensors in Heihe for matching a footprint of MODIS observations. But it is generally impossible to operationally maintain a very-high density of networks for large-scale areas under the constraints of financial supports.

Li, X., Cheng, G., Liu, S., Xiao, Q., Ma, M., Jin, R., Che, T., Liu, Q., Wang, W., Qi, Y., Wen, J., Li, H., Zhu, G., Guo, J., Ran, Y., Wang, S., Zhu, Z., Zhou, J., Hu, X., and Xu, Z.: Heihe Watershed Allied Telemetry Experimental Research (HiWATER), 16, 2013.

**Comments #4.** Figure 2, which Kriging method was used for TR-4?

**Responses #4**: Thanks for the comment. We used the FRK method to model the site residuals

from RF models constructed for TR-4. We have modified Figure 2 (Line 195).

**Comments #5.** Figure 4, why are RMSE and MAE so large in TR-1 and TR-6?

**Responses #5**: Thanks for your important comments. In general, SAT estimation models developed for regions with complex geographical environments and climatic dynamics such as polar regions (Nielsen et al., 2023; Meyer et al., 2016; Kilibarda et al., 2014) exhibit larger predictive errors and uncertainties. There are very limited ground stations in the polar regions (TR-1 and TR-6) and sampling representativeness by the scarce stations will deteriorate the predictive performance of SAT estimation models trained using samples from the stations in the regions. In our revised manuscript (Lines 360-371), we have discussed issue of scarcity of stations for SAT model building, and state that developing models based on transfer learning (Wang W. et al., 2025) offers a promising and important approach for estimating SAT in regions with the scarcity of stations (Lines 370-371).

Kilibarda, M., Hengl, T., Heuvelink, G.B.M., Gräler, B., Pebesma, E., Perčec Tadić, M., Bajat, B., 2014. Spatio-temporal interpolation of daily temperatures for global land areas at 1 km resolution. J. Geophys. Res.: Atmos. 119, 2294–2313. https://doi.org/10.1002/2013JD020803

Nielsen, E. B., Katurji, M., Zawar-Reza, P., and Meyer, H.: Antarctic daily mesoscale air temperature dataset derived from MODIS land and ice surface temperature, Sci Data, 10, 833, https://doi.org/10.1038/s41597-023-02720-z, 2023.

Meyer, H., Katurji, M., Appelhans, T., Müller, M., Nauss, T., Roudier, P., and Zawar-Reza, P.: Mapping Daily Air Temperature for Antarctica Based on MODIS LST, Remote Sens., 8, 732, https://doi.org/10.3390/rs8090732, 2016.

Wang, W., Brönnimann, S., Zhou, J., Li, S., and Wang, Z.: Near-surface air temperature estimation for areas with sparse observations based on transfer learning, ISPRS Journal of Photogrammetry and Remote Sensing, 220, 712–727, https://doi.org/10.1016/j.isprsjprs.2025.01.021, 2025.

**Comments #6.** Figure 5, why does the RMSE of the model have such strong periodicity?

**Responses #6**: Thanks for your important comments. The changing of surface air temperature (SAT) has very strong seasonality and periodicity. For the northern hemisphere, SAT reaches two extremes for summer months and winter months. In general, it is harder to build models for extreme SAT (summer or wither months) than for moderate conditions of SAT (spring and autumn months). Therefore, the RMSE for models developed across different months exhibit periodicity. The periodical variability of RMSE for SAT models has also been reported in previous studies (Wang M. et al., 2024; Yao R. et al, 2023). We have discussed the issue in Line 390-410.

Yao, R., Wang, L., Huang, X., Cao, Q., Wei, J., He, P., Wang, S., and Wang, L.: Global seamless and high-resolution temperature dataset (GSHTD), 2001–2020, Remote Sens. Environ., 286, 113422, https://doi.org/10.1016/j.rse.2022.113422, 2023.

Wang, M., Wei, J., Wang, X., Luan, Q., and Xu, X.: Reconstruction of all-sky daily air temperature datasets with high accuracy in China from 2003 to 2022, Sci. Data, 11, 1133, https://doi.org/10.1038/s41597-024-03980-z, 2024.

**Comments #7.** Line 211, what is RF-KR?

**Responses #7**: Thanks for your concerns. We developed hybrid estimation models that integrate RF and residual kriging for reconstructing our dataset. Two types of kriging including OK and FRK have been utilized for constructing the hybrid models. In our manuscript, we use the abbreviation RF-KR to generally refer the hybrid models. We have reorganized and rewritten some contents of section 3.2 (Lines 215-223, and Lines 244-251) to clearly describe all model abbreviations used in our manuscript.

**Comments #8.** Line 226, Should the formula number be Eq. (3)?

**Responses #8**: Thank you for pointing out this mistake. We have corrected it (Line 261).

**Comments #9.** Is the time label of the dataset local time or UTC? This is critical for users.

**Responses #9**: Thanks for this important comment. The time standard for our GHRSAT product is UTC. We have added contents for the time standard in sec. 3.1 (Lines 199-200) and sec. 5 (Line 676).

**Comments #10.** For the air temperature estimate in the case of sparse sites, please refer to these two articles: https://doi.org/10.1016/j.isprsjprs.2025.01.021;https://doi.org/10.1109/JSTARS.2022.3161800

**Responses #10**: Thanks for your valuable comments. The two articles are very pertaining to our study, and we have referred the two articles in our revised manuscript to discuss the significance of building models based on reconstructed LST (Wang et al. 2022, in Line 85) and state the significance of developing SAT model based on transfer learning for estimating SAT in regions with limited stations (Wang et al., 2025, in Line 371).

---

## Author Comment (AC3)

**Responses to RC3**

*Dear Reviewer #3:*

*Thanks very much for your time on reviewing our manuscript. We sincerely thank the reviewer for your efforts on the reviewing of our manuscript. We deeply appreciate your valuable comments on our manuscript, and we have carefully revised the manuscript according to the comments. The point-by-point responses to your comments are provided in this document.*

*Best regards,*
*Zhenwei Zhang*
*Nanjing University of Information Science & Technology*

**Comments #1.** Imbalance in Model Description: The description of the model in the methods section is unbalanced, with much more emphasis on Kriging compared to Random Forest (RF).

**Responses #1**: Thanks for your valuable comments on our manuscript. We have revised Sec. 3.2 to supplement some information about the advantages of RF, and more importantly restructured the section to more clearly describe how the RF models are integrated with two kriging techniques to meet the tasks of constructing hourly SAT estimation models in our study (Lines 234-237). In addition, there are two types of residual kriging techniques (OK and FRK) have been integrated into the SAT estimation models, which is described in Sec. 3.2. Therefore, the section contains more contents relevant to the two kriging methods than RF. But the content length for each kriging method is roughly the same as that for RF.

**Comments #2.** Handling Missing Data: The paper mentions removing records with poor quality in the ground station data but does not explain in detail how missing data is handled, especially missing data with different temporal and spatial resolutions. If there is a significant amount of missing data, the model's accuracy and generalizability may be impacted

**Responses #2**: Thanks for your important comments. Thanks for your important comments. There are complete and detailed official data documentation for the ISD observational dataset (see https://www.ncei.noaa.gov/data/global-hourly/doc/). For modeling SAT in our study, we used the station records from ISD without any quality-control issues, specifically the records passed all quality control checks (see page 10 of the isd-format-document.pdf), which has been pointed out in our revised manuscript (Line 206 and Lines 208-210). In fact, there is only a slight portion of records with missing SAT data or with QA issues. Even without any missing records, the current ground observational data is inadequate for use in the studies on SAT estimation and other applications. In general, there are high-coverage and high-density ground observational networks in developed countries and regions, such as America and

Europe, in contrast, the poor regions (for example, Africa) and polar areas have very limited coverage of ground stations. Estimation models for SAT will be more representative with higher accuracy when training using samples from high-coverage and high-quality networks of stations. However, there will be a long road ahead to establish such station networks, especially for undeveloped regions. The importance of ground station data for building SAT estimation models and the limitations of our models by ground station data have been discussed in Sec. 4.5 in our revised manuscript (Line 644-657).

**Comments #3.** Selection of Covariates: The paper uses multiple spatial covariates such as NDVI, elevation, latitude/longitude, and hour of the day, but it does not provide a detailed discussion of the rationale behind selecting these covariates or their applicability in different regions.

**Responses #3**: Thanks for your valuable comments. The primary fundamental of selecting the input variables for SAT estimation models is by considering whether incorporating the covariates into SAT estimation models will contribute the predictive performance of the models. As our study aimed at building estimation models for global land areas, it is inevitable **to only consider the covariates for which datasets are available at the global scale** in the time period 2011-2023. There are some differences in the selection of covariates for SAT estimation among previous studies, which is **primary due to the localized consideration of modeling SAT for specific study areas and the constraints of data availability**. For examples, previous studies have developed SAT estimation models considering covariates for satellite-based snow cover (Wang W. et al., 2025) and surface structural properties derived from lidar data (Venter Z. S., et al., 2020). However, the models utilizing these covariates are only restricted to the study areas that the studies focused on, and cannot be generalized to other regions due to data unavailability for these covariates in other regions.

**We should note that there are very limited input auxiliary covariates that available at the global scale for use in modeling hourly SAT**. The auxiliary variables used in our study have been widely used in previous studies for building SAT estimation models. More importantly, data for the auxiliary variables used in our study are available at the global scale, and can be easily and publicly accessed online. In our exploratory experiments for building SAT models based on RF for different task regions using the selected covariates, we find that models with all the selected covariates exhibited better validation performance than models considering only subsets of the covariates. To more clearly state our consideration for selecting the auxiliary variables for modeling hourly SAT in our study, we have rewritten some parts of Sec. 2.3 (see Lines 155-166).

Wang, W., Brönnimann, S., Zhou, J., Li, S., and Wang, Z.: Near-surface air temperature estimation for areas with sparse observations based on transfer learning, ISPRS Journal of Photogrammetry and Remote Sensing, 220, 712–727, https://doi.org/10.1016/j.isprsjprs.2025.01.021, 2025.

*Venter, Z. S., Brousse, O., Esau, I., and Meier, F.: Hyperlocal mapping of urban air temperature using remote sensing and crowdsourced weather data, Remote Sens. Environ., 242, 111791, https://doi.org/10.1016/j.rse.2020.111791, 2020.*

**Comments #4.** Resampling of NDVI and Elevation Data: The resampling method for NDVI and elevation data is not clearly stated, which could affect data quality.

**Responses #4**: Thanks for your important suggestions. We agree with you that the resampling of various satellite-based and geoscientific data is very important and has impacts on data quality. We used the elliptical weighted averaging (EWA) method, which is stated Lines 201-202 of our revised manuscript. The EWA method is widely used for regridding MODIS data. More information about the method could be refereed to https://pyresample.readthedocs.io

**Comments #5.** Limited Model Performance Evaluation: The paper only uses RMSE and MAE as performance metrics, lacking analysis of systemic bias (Bias) or coefficient of determination ($R^2$), which makes it difficult to fully assess the model's performance.

**Responses #5**: Thanks for your important comments. We agree with you that the diverse metrics will help to fully assess the estimation models used for reconstructing the GHRSAT datasets. In addition to the RMSE for our validated models, we have provided the comparison of our models in terms of the performance metrics including Bias, MAE and coefficient of determination (see the revised supplement for our manuscript). We have additionally performed site-based cross-validation (CV) for all models developed in our study (see Lines 289-294, Sec. 3.3). The validation results for sited-based CV have been discussed in our revised manuscript (Lines 26-29 in the Abstract, Lines 332-359 in Sec. 4.1, and Lines 694-699 in Sec. 6). Fig. 4 has been revised to contain the overall validation results for our models under both sample-based and site-based cross-validation (Lines 385-390).

**Comments #6.** Discussion on Practical Application: The discussion section could benefit from further elaboration on how the research results could be applied to real-world problems. Additionally, the limitations of the current study should be clearly stated, along with potential directions for future improvements.

**Responses #6**: Thanks for your valuable comments. In our revised manuscript, we have added Sec. 4.5 (Lines 643-657) to clearly discuss the limitations, practical applications and potential improvements of our study. We discussed the limitations of our study in modeling of hourly SAT from the aspects of methodology and data sources. The potential applications of our reconstructions for the field of remotely sensed estimation, and applications of our reconstructed GHRSAT dataset have been briefly discussed.